# DiTFastAttn: Attention Compression for Diffusion Transformer Models

**Zhihang Yuan**[*1,2] **Hanling Zhang**[*1,2] **Pu Lu**[*1] **Xuefei Ning**[1 †] **Linfeng Zhang**[3]

**Tianchen Zhao**[1,2] **Shengen Yan**[2] **Guohao Dai**[3,2] **Yu Wang**[1]

[1]Tsinghua University [2]Infinigence AI [3]Shanghai Jiao Tong University

## Abstract

Diffusion Transformers (DiT) excel at image and video generation but face computational challenges due to the quadratic complexity of self-attention operators. We propose DiTFastAttn, a post-training compression method to alleviate the computational bottleneck of DiT. We identify three key redundancies in the attention computation during DiT inference: (1) spatial redundancy, where many attention heads focus on local information; (2) temporal redundancy, with high similarity between the attention outputs of neighboring steps; (3) conditional redundancy, where conditional and unconditional inferences exhibit significant similarity. We propose three techniques to reduce these redundancies: (1) *Window Attention with Residual Sharing* to reduce spatial redundancy; (2) *Attention Sharing across Timesteps* to exploit the similarity between steps; (3) *Attention Sharing across CFG* to skip redundant computations during conditional generation. We apply DiTFastAttn to DiT, PixArt-Sigma for image generation tasks, and OpenSora for video generation tasks. Our results show that for image generation, our method reduces up to 76% of the attention FLOPs and achieves up to $1.8\times$ end-to-end speedup at high-resolution (2k $\times$ 2k) generation.

## 1 Introduction

Recently, diffusion transformers (DiT) have gained increasing popularity in image (Peebles & Xie, 2023; Chen et al., 2024) and video generation (Brooks et al., 2024). However, a major challenge with DiTs is their substantial computational demand, particularly noticeable when generating high-resolution content. On the one hand, traditional transformer architectures, with their self-attention mechanism, have an $\mathcal{O}(L^2)$ complexity to the input token length $L$. This quadratic complexity leads to significant computational costs as the resolution of images and videos escalates. As demonstrated in Fig. 1, the attention computation becomes the primary computational bottleneck during the inference process as image resolution increases. Specifically, if a $2K \times 2K$ image is tokenized into 16k tokens (Chen et al., 2024), this requires several seconds for attention computation, even on high-end GPUs such as the Nvidia A100. On the other hand, the inference process of diffusion requires a substantial number of neural network inferences due to the multiple denoising steps and the classifier-free guidance (CFG) technique (Ho & Salimans, 2022).

Previous efforts to accelerate attention mechanisms, such as Local Attention, Swin Transformer (Liu et al., 2021), and Group Query Attention (GQA) (Ainslie et al., 2023), mainly focused on designing the attention mechanism or network architecture. While effective in reducing computational costs, these approaches necessitate large retraining costs. Due to the substantial data and computational requirements for training a DiT, there is a need for post-training compression methods.

In this work, we identify three types of redundancy in the attention computation of DiT inference and propose a post-training model compression method, DiTFastAttn, to address these redundancies:

---

[*]Equal contribution. [†] Corresponding author and project advisor: `foxdoraame@gmail.com`. Project Website: `http://nics-effalg.com/DiTFastAttn`.

38th Conference on Neural Information Processing Systems (NeurIPS 2024).

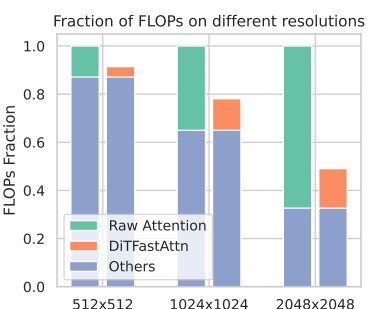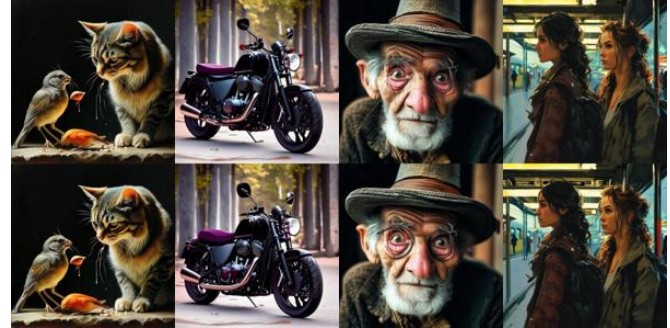

Figure 1: **Left**: The efficiency benefits of applying DiTFastAttn on PixArt-Sigma (Chen et al., 2024) when generating images of different resolutions. The Y-axis shows the #FLOPs fraction normalized by the #FLOPs of the original model. **Right**: The qualitative results of applying DiTFastAttn on 1024×1024 PixArt-Sigma.

(1) *Redundancy in the spatial dimension*. Many attention heads primarily capture local spatial information, with attention values for distant tokens nearing zero. To reduce the redundancy, we opt to use window attention instead of full attention for certain layers. However, directly discarding all attention computation outside the window leads to significant performance degradation. To maintain the performance in a training-free way, we propose to cache the residual between the outputs of full and window attention at one step and reuse this residual for several subsequent steps. We refer to this technique as **Window Attention with Residual Sharing (WA-RS)**.

(2) *Similarity between the neighboring steps in attention outputs*. The attention outputs of the same attention head across neighboring steps can be highly similar. We propose the **Attention Sharing across Timesteps (AST)** technique that exploits this step-wise similarity to accelerate attention computation.

(3) *Similarity between the conditional and unconditional inference in attention outputs*. We observe that in CFG, the attention outputs of conditional and unconditional inference exhibit significant similarity (SSIM $\geq$ 0.95) for certain heads and timesteps. Therefore, we propose the **Attention Sharing across CFG (ASC)** technique to skip redundant computation during unconditional generation.

We conduct extensive experiments to evaluate DiTFastAttn using multiple DiT models, including DiT-XL (Peebles & Xie, 2023) and PixArt-Sigma (Chen et al., 2024) for image generation, and Open-Sora (Open-Sora, 2024) for video generation. Our findings demonstrate that DiTFastAttn consistently reduces the computational cost. Notably, the higher the resolution, the greater the savings in computation and latency. For instance, with PixArt-Sigma, DiTFastAttn delivers a 20% to 76% reduction in attention computation and a end-to-end speedup of up to 1.8× during the generation of 2048×2048 images.

## 2   Related Work

### 2.1   Diffusion Models

Diffusion models (Ho et al., 2020; Rombach et al., 2022; Peebles & Xie, 2023; Chen et al., 2024; Brooks et al., 2024) have gained significant attention due to their superior generative performance compared to GANs (Creswell et al., 2018). Early diffusion models (Ho et al., 2020; Rombach et al., 2022) are implemented based on the U-Net architecture. To achieve better scalability, DiT (Peebles & Xie, 2023) utilizes the transformer architecture instead of U-Net. Diffusion transformer is applied in the fields of image and video generation. PixArt-Sigma (Chen et al., 2024) demonstrates the diffusion transformer's capability to generate high-resolution images up to 4K. Sora (Brooks et al., 2024) presents the diffusion transformer's ability to generate videos.

## 2.2 Vision Transformer Compression

The computational overhead of attention has garnered significant attention. FlashAttention (Dao, 2023) divides the input tokens into smaller tiles to minimize redundant memory accesses and optimize latency. Some studies highlight the quadratic complexity of attention computation and improve efficiency through token pruning, achieved by filtering (Rao et al., 2021; Liu et al., 2022; Wu et al., 2023) or merging (Lu et al., 2023; Huang et al., 2023; Wu et al., 2023) tokens at different stages of the network. DynamicViT (Rao et al., 2021) employs a prediction network to dynamically filter tokens. Adaptive Sparse ViT (Liu et al., 2022) filters tokens by simultaneously considering the attention values and the L2 norm of the features. Lu et al. (2023) trains a network with segmentation labels to direct the merging operations of tokens in regions with similar content. Huang et al. (2023) conducts attention computations after downsampling tokens and subsequently upsampling to recover the spatial resolution. Wu et al. (2023) demonstrates that deeper layers are more suitable for filtering tokens, whereas shallower layers are more appropriate for merging tokens.

## 2.3 Local Attention

Various studies have delved into the utilization of local attention patterns, where each token attends to a set of neighboring tokens within a fixed window size, aiming to mitigate the computational burden associated with processing long sequences. The concept of local windowed attention was initially introduced by Beltagy et al. (2020) in Longformer, presenting an attention mechanism that scales linearly with sequence length. Bigbird (Zaheer et al., 2020) extends this idea by incorporating window attention, random attention, and global attention mechanisms, enabling the retention of long-range dependencies while mitigating computational costs. In the realm of computer vision, Swin Transformer (Liu et al., 2021) adopts a similar approach by confining attention computation to non-overlapping local windows, utilizing shifted windows across different layers to capture global context efficiently. Twins Transformer(Chu et al., 2021), FasterViT(Vasu et al., 2023), and Neighborhood attention transformer (Hassani et al., 2023) employ window-based attention to enhance computational efficiency, leveraging different module designs such as global sub-sampled attention and hierarchical attention to exploit global context effectively. In our work, we employ fixed-sized window attention to accelerate pretrained Diffusion Transformer models and introduce a novel technique named Window Attention with Residual Sharing to preserve long-range dependencies for image tokens.

## 2.4 Attention Sharing

GQA (Ainslie et al., 2023) divides query heads into $G$ groups. Each query retains its own parameters, while each group shares a key and value, reducing memory usage and improving efficiency. PSVIT (Chen et al., 2021) shows that attention maps between different layers in ViT have significant similarity and suggests sharing attention maps across layers to reduce redundant computation. Deepcache (Ma et al., 2023) demonstrates that high-level features in U-Net framework diffusion models are similar across timesteps. Deepcache proposes reusing U-Net's high-level features and skipping intermediate layers' computation to accelerate denoising process. TGATE (Zhang et al., 2024) shows that the cross-attention output of text-conditional diffusion models converges to a fixed point after several denoising timesteps. TGATE caches this output once it converges and keeps it fixed during the remaining denoising steps to reduce computational cost. In DiTFastAttn, we demonstrate the similarity of attention outputs both CFG-wise and step-wise. We also consider the differences in similarity across different layers at various steps to share attention outputs CFG-wise and step-wise.

## 2.5 Other Methods to Accelerate Diffusion Models

Network quantization is a widely used technique for reducing the bitwidth of weights and activations, effectively compressing both image generation models (Shang et al., 2023; Zhao et al., 2024b) and video generation models (Zhao et al., 2024a). Scheduler optimization is another popular approach aimed at decreasing the number of timesteps in the denoising process (Song et al., 2020; Lu et al., 2022; Liu et al., 2023a). Additionally, distillation serves as an effective method for minimizing the timesteps required during denoising (Salimans & Ho, 2022; Meng et al., 2023; Liu et al., 2023b). DiTFastAttn offers a complementary solution, as it operates independently of the specific quantization bitwidth, scheduler, and timesteps employed.

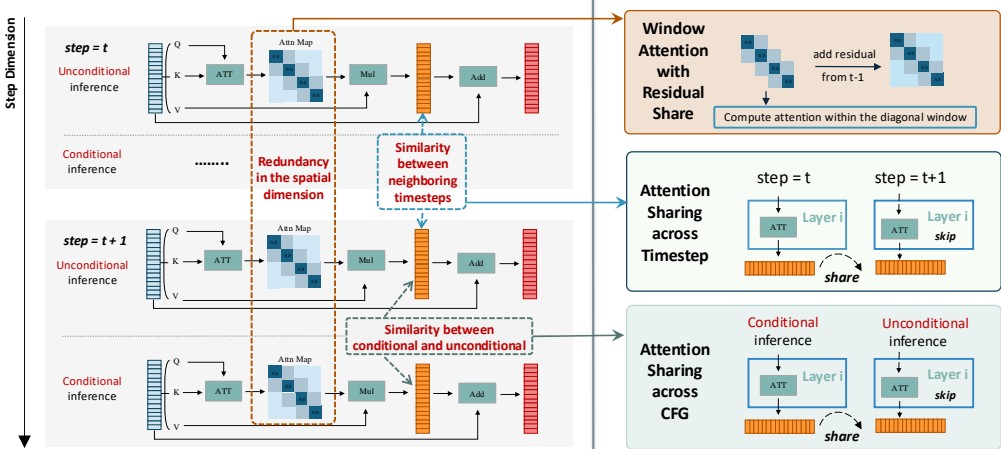

Figure 2: **Types of Redundancy and Corresponding Compression Techniques**. **Left:** Redundancy in the spatial dimension, denoising steps, and CFG. **Right:** Techniques implemented in DiTFastAttn to reduce redundancy for each type. DiTFastAttn employs window attention to minimize attention redundancy, while maintaining performance using residuals. Additionally, attention outputs are shared both step-wise and CFG-wise to reduce redundancy.

## 3  Method

### 3.1  Overview

In this section, we demonstrate the redundancy in the inference process of diffusion models with transformers. In the denoising process, we identify three types of redundancy, as shown in Fig. 2: (1) *Redundancy in the spatial dimension.* (2) *Similarity between the neighboring steps in attention outputs.* (3) *Similarity between the conditional and unconditional inference in attention outputs.* To address these redundancies, we propose three compression techniques, as shown in Fig. 2: (1) In Sec. 3.2, we introduce **Window Attention with Residual Sharing** to reduce spatial redundancy. (2) In Sec. 3.3, we introduce **Attention Sharing across Timesteps** to exploit step-wise similarities, thereby enhancing model efficiency. (3) In Sec. 3.4, we introduce **Attention Sharing across CFG** to reduce redundancy by utilizing similarity between conditional and unconditional generation. In Sec. 3.5, we introduce a simple greedy method to decide the compression plan, i.e., select the appropriate compression technique for each layer and step.

### 3.2  Window Attention with Residual Sharing (WA-RS)

We can observe the spatial locality of attention in many transformer layers in pre-trained DiTs. As shown in Fig. 3(a), attention values concentrate within a window along the diagonal region of the attention matrix. Therefore, replacing full attention with fixed-size window attention for some layers can preserve most of the values in the attention matrix during inference. By computing attention values only within a specified window, the computation cost of attention can be largely reduced.

However, some tokens still attend to a small set of spatial distant tokens. Discarding these dependencies negatively affects model performance. Mitigating this issue using only window attention necessitates a large window size to capture these dependencies. Consequently, this approach achieves minimal reduction in computational cost, thereby hindering acceleration efforts.

**Cache and Reuse the Residual for Window Attention.** To address the aforementioned issue, we investigate the information loss caused by using window attention. As shown in Fig. 3(a), the residual between the outputs of full and window attention exhibits a small variation across steps, unlike the output of window attention. This observation motivates us to cache the residual of window attention and full attention in one step and reuse it in subsequent steps.

Fig. 3(b) illustrates the computation of WA-RS: at each step, for each window attention layer, we compute the window attention and add a residual cached from the previous step to the output. We

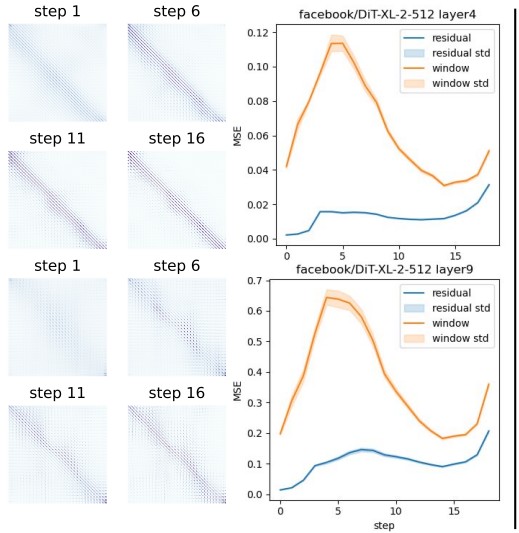
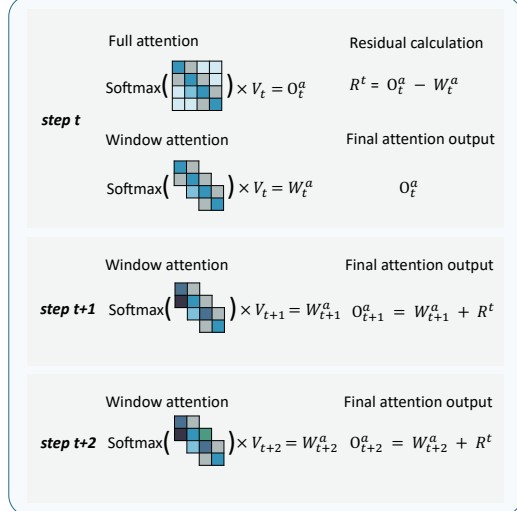

(a) Left: Examples of attention map that shows sliding window pattern. Right: MSE of proceeding and current step attention

(b) Illustration of Window Attention with Residual Sharing

Figure 3: **Window Attention with Residual Sharing**. (a) **Left**: Example of the attention map showing the window pattern. **Right**: The MSE between the window attention outputs in the previous and current step (yellow line) versus the MSE between the output residuals of window and full attention in the previous and current step (blue line). The output residual exhibits minimal changes over the steps. (b) Computation of Window Attention with Residual Sharing. Window attention that illustrates significant changes is recalculated. Residuals that change minimally are cached and reused in subsequent steps.

denote the set of steps that share the residual value $\mathbf{R}_r$ as $\mathbf{K}$, the full attention at step $r$ as $\mathbf{O}_r$, the window attention at step $k$ as $\mathbf{W}_k$. For the first step in the set $r = \min(\mathbf{K})$, the computation of WA-RS goes as follows:

$$
\begin{aligned}
\mathbf{O}_r &= \text{Attention}(\mathbf{Q}_r, \mathbf{K}_r, \mathbf{V}_r), \\
\mathbf{W}_r &= \text{WindowAttention}(\mathbf{Q}_r, \mathbf{K}_r, \mathbf{V}_r), \\
\mathbf{R}_r &= \mathbf{O}_r - \mathbf{W}_r.
\end{aligned}
\tag{1}
$$

For a subsequent step in the set $k \in \mathbf{K}$, the computation of WA-RS goes as follows:

$$
\begin{aligned}
\mathbf{W}_k &= \text{WindowAttention}(\mathbf{Q}_k, \mathbf{K}_k, \mathbf{V}_k), \\
\mathbf{O}_k &= \mathbf{W}_k + \mathbf{R}_r.
\end{aligned}
\tag{2}
$$

### 3.3 Attention Sharing across Timesteps (AST)

The sequential nature of the denoising process in diffusion models is a major bottleneck for inference speed. Here, we compare the attention outputs at different steps during the denoising process. We find that for some layers, the attention outputs at certain steps show significant similarity to those of adjacent steps. Fig. 4(a) presents the cosine similarity between the attention outputs at different steps. We can draw two primary observations: (1) There is a noticeable temporal similarity between the attention outputs; (2) This similarity varies across steps and layers.

To exploit this similarity to reduce the computational cost, we propose the AST technique. Specifically, for a set of steps with their attention outputs similar to each other, we cache the earliest step's attention output $\mathbf{O}$ and reuse it, thereby skipping the computation at the subsequent steps.

### 3.4 Attention Sharing across CFG (ASC)

Classifier-free guidance (CFG) is widely used for conditional generation (Ho & Salimans, 2022; Ramesh et al., 2022; Saharia et al., 2022). In each step of the inference process for conditional

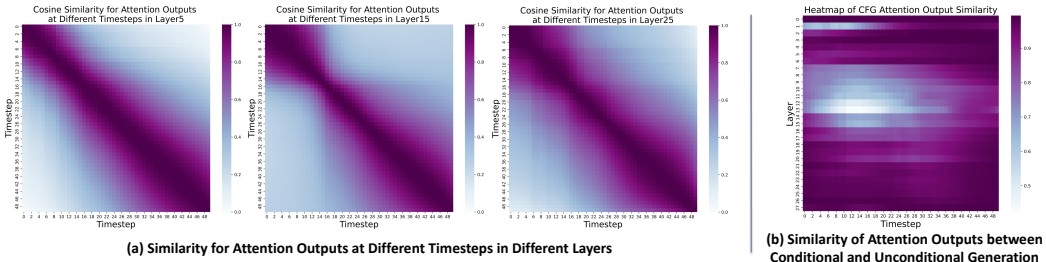

**(a) Similarity for Attention Outputs at Different Timesteps in Different Layers**

**(b) Similarity of Attention Outputs between Conditional and Unconditional Generation**

Figure 4: **Similarity of Attention Outputs Across Step and CFG Dimensions in DiT.** (a) Similarity of attention outputs across step dimension in different layers. (b) Similarity between conditional and unconditional attention outputs in various layers at different steps

generation, CFG performs two neural network inferences: one with the conditional input and one without. This doubles the computational cost compared with unconditional generation. As shown in Fig. 4(b), for many layers and steps, the similarity between the attention outputs in the conditional and unconditional neural network evaluations is high.

Based on this observation, we propose the ASC technique that reuses the attention output from the conditional neural network evaluation in the unconditional neural network evaluation.

## 3.5 Method for Deciding the Compression Plan

The aforementioned techniques, including WA-RS, AST, and ASC, can effectively cut down the computational cost while maintaining the performance. As shown in Fig. 3 and Fig. 4, different layers have different redundancies in different time steps. Therefore, it is crucial to properly decide the compression plan, i.e., which techniques should be applied for each layer at each step.

We develop a simple greedy method to select the appropriate strategy (a combination of techniques) from a strategy list $\mathcal{S}$ =[AST, WA-RS + ASC, WA-RS, ASC] for each step and each layer. As shown in Alg. 1, we determine the strategies step by step and layer by layer. For each step and transformer layer, we apply each of the four compression strategies and calculate the loss between the model outputs with and without compression for the current step, $L(O, O')$. Then, we select the strategy with the highest computation reduction ratio with loss below a threshold $\frac{i}{|M|}\delta$, where $i$ is the layer index and $|M|$ is the number of layers in the model. If none of the four strategies meet the threshold, we do not apply compression for this layer at that step.

---

**Algorithm 1:** Method for Deciding the Compression Plan

**Input**  : Transformer Model $M$, Total Step $T$, Compression Strategy List $\mathcal{S}$, Threshold $\delta$
**Output** : dictionary `dict` that stores selected compression techniques
Initialize `dict`
**for** step $t$ in $T$ **do**
  $O \leftarrow$ compute the output of the uncompressed $M$
  **for** transformer layer $i$ in $M$ **do**
    **for** $m \in \mathcal{S}$ order by ascending compression ratio **do**
      compress layer $i$ in step $t$ using compression strategy $m$
      $O' \leftarrow$ compute the output of $M$
      **if** $L(O, O') < \frac{i}{|M|}\delta$ **then**
        update $m$ as the selected strategy of layer $i$ and step $t$ in `dict`
        break
**return** `dict`

---

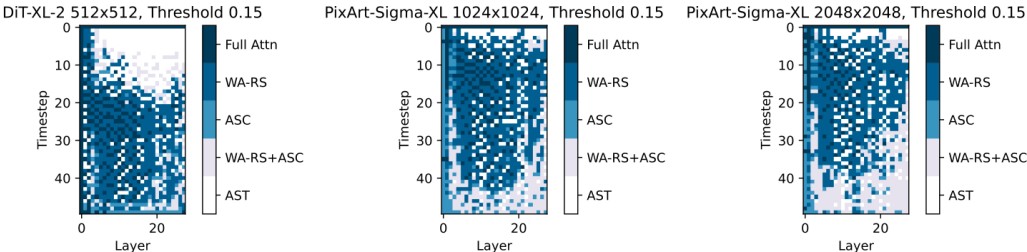

Figure 5: Compression plan for DiT-XL-512, PixArt-Sigma-XL-1024 and PixArt-Sigma-XL-2K at D6 with the number of DPM-Solver steps set to 50.

## 4    Experiments

### 4.1    Settings

We evaluate DiTFastAttn on three commonly used diffusion transformers: DiT (Peebles & Xie, 2023) and Pixart-Sigma (Chen et al., 2024) for image generation tasks, and Open-Sora (Open-Sora, 2024) for video generation tasks. To demonstrate compatibility with fast sampling methods, we build our method upon 50-step DPM-Solver for DiT and Pixart-Sigma, and 200-step IDDPM (Nichol & Dhariwal, 2021) for Open-Sora.

For calculating quality metrics, we use ImageNet as the evaluation dataset for DiT and MS-COCO as the evaluation dataset for PixArt-Sigma. MS-COCO 2014 caption is used as text prompt for Pixart-Sigma models' image generation. To evaluate generation quality, we generate 50k images for DiT models and 30k images for PixArt-Sigma models. Following previous studies, we employ FID (Heusel et al., 2017), IS (Salimans et al., 2016) and CLIP score (Hessel et al., 2021) as the evaluation metrics. We measure the latency per sample on a single Nvidia A100 GPU.

We use mean relative absolute error for $L(O, O')$ and experiment with and different thresholds $\delta$ at intervals of 0.025. We denote these threshold settings as D1 ($\delta$=0.025), D2 ($\delta$=0.05), ..., D6 ($\delta$=0.15), respectively. We set the window size of WA-RS to 1/8 of the token size.

### 4.2    Results on Image Generation

**Results of Evaluation Metrics and Attention FLOPs.** DiTFastAttn is applied to the pre-trained DiT-XL-2-512, PixArt-Sigma-1024, and PixArt-Sigma-2K models. Table 1 displays the evaluation results of these models. For the DiT-XL-2-512 and PixArt-Sigma-1024 models, configurations D1, D2, and D3 nearly match the performance of the original models in terms of IS and FID metrics. Comparison of compression effects and evaluation metrics between the three models reveals that as image resolution increases, DiTFastAttn not only achieves greater compression but also better preserves the generative performance of the models.

**Compression Plan after Search.** Fig. 5 illustrates the compression plan under the D6 setting. For the DiT model, AST and ASC are utilized in the early timesteps, with full attention primarily appearing in the initial attention layers. In contrast, the PixArt-Sigma model employs AST sporadically in the first two layers and in the middle attention layers during the intermediate timesteps, while the combination of WA-RS and ASC is notably predominant in the final timesteps. This variability in the distribution of different types of redundancy across models highlights the absence of a universal compression strategy, underscoring the necessity for tailored plan searches. Additional compression plans for other settings are provided in A.5.

**Visualization of DiTFastAttn's Generation Results.** Fig. 6 presents image generation samples from DiTFastAttn. The D1, D2, and D3 configurations of the DiT-XL-2-512 and PixArt-Sigma-1024 models demonstrate visual generation quality comparable to the original models. In contrast, D4, D5, and D6 achieve greater compression, exhibiting slight variations in detail while still producing acceptable-quality images. The PixArt-Sigma-2K model maintains image quality similar to the original up to D4, with D5 and D6 also generating high-quality outputs. This suggests that our compression method effectively preserves generation quality, even when reducing attention computation by over 50% and compressing to 33% at higher resolutions.

Table 1: Image generation performance of DiTFastAttn at various image resolutions under various compression ratios. The FID, IS, and CLIP results are marked in different makers. The 'Attn FLOPs' represents the fraction of computation in the multi-head attention module compared to the raw model.

| Model | DiT-XL-2 512×512 | | | PixArt-Sigma-XL 1024×1024 | | | | PixArt-Sigma-XL 2048×2048 | | | |
|---|---|---|---|---|---|---|---|---|---|---|---|
| Score | IS | FID | Attn FLOPs | IS | FID | CLIP | Attn FLOPs | IS | FID | CLIP | Attn FLOPs |
| Raw | 408.16 | 25.43 | 100% | 24.33 | 55.65 | 31.27 | 100% | 23.67 | 51.89 | 31.47 | 100% |
| D1 | 412.24 | 25.32 | 85% | 24.27 | 55.73 | 31.27 | 90% | 23.28 | 52.34 | 31.46 | 81% |
| D2 | 412.18 | 24.67 | 69% | 24.25 | 55.69 | 31.26 | 74% | 22.90 | 53.01 | 31.32 | 60% |
| D3 | 411.74 | 23.76 | 59% | 24.16 | 55.61 | 31.25 | 63% | 22.96 | 52.54 | 31.36 | 46% |
| D4 | 391.80 | 21.52 | 49% | 24.07 | 55.32 | 31.24 | 52% | 22.95 | 51.74 | 31.39 | 36% |
| D5 | 370.07 | 19.32 | 41% | 24.17 | 54.54 | 31.22 | 44% | 22.82 | 51.21 | 31.34 | 29% |
| D6 | 352.20 | 16.80 | 34% | 23.94 | 52.73 | 31.18 | 37% | 22.38 | 49.34 | 31.28 | 24% |

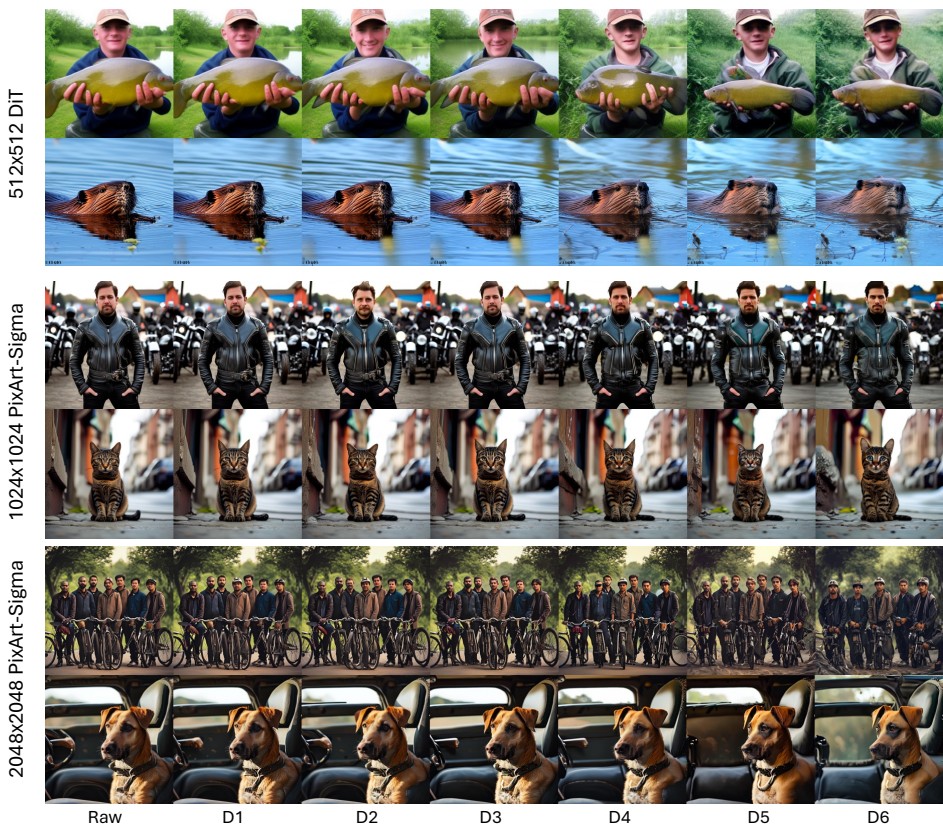

Figure 6: Image generation samples at various image resolutions under various compression ratios.

## 4.3 Results on Video Generation

We apply DitFastAttn on OpenSora for video generation with thresholds from 0.01 to 0.06. The results are shown in Fig. 7. Specifically, the reduction in attention computation for these configurations are as follows: 11.68%, 27.63%, 40.48%, 48.98%, 50.75%, 55.76% respectively. Latency at raw setting is 35.79 seconds where as the performance with the aforementioned configurations were as follows: 34.79 seconds, 33.39 seconds, 31.74 seconds, 31.29 seconds, 31.29 seconds, and 31.17 seconds, achieving 1.02× to 1.15× end-to-end speedup. For an extended analysis, additional results are provided in A.3.

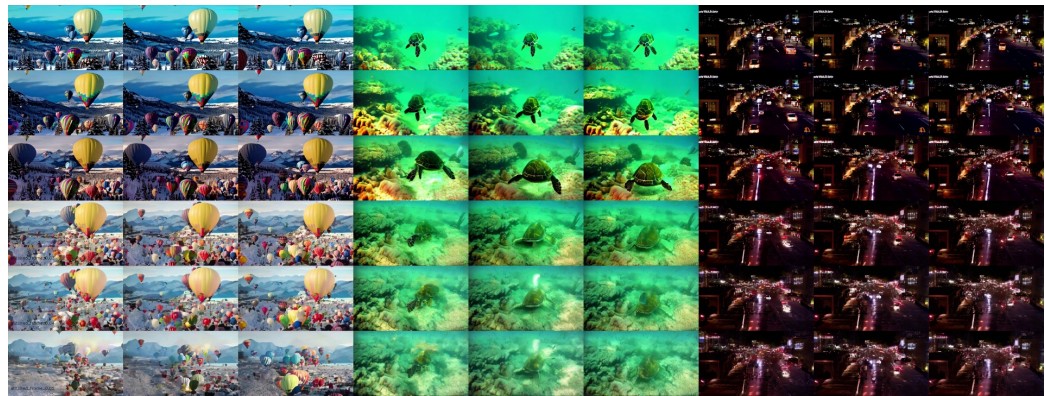

Figure 7: Comparison of video generation using OpenSora V1.1 at 240p resolution with 16 frames.

Table 2: FLOPs fraction and latency fraction of DitFastAttn in Diffusion Transformers comparing with original attention. The latency is evaluated on the Nvidia A100 GPU.

| Model | Seqlen | Metric | ASC | WA-RS | WA-RS+ASC | AST |
|---|---|---|---|---|---|---|
| DiT-XL-2 512×512 | 1024 | Attn FLOPs | 50% | 77% | 38% | 0% |
| | | Attn Latency | 59% | 85% | 51% | 4% |
| PixArt-Sigma-XL 1024×1024 | 4096 | Attn FLOPs | 50% | 51% | 26% | 0% |
| | | Attn Latency | 54% | 54% | 31% | 3% |
| PixArt-Sigma-XL 2048×2048 | 16384 | Attn FLOPs | 50% | 33% | 16% | 0% |
| | | Attn Latency | 52% | 35% | 19% | 1% |

## 4.4 #FLOPs Reduction and Speedup

**Compression Results of DiTFastAttn on Various Sequence Lengths.** We implement DiTFastAttn based on FlashAttention-2 (Dao, 2023). Table 2 shows the FLOPs fraction and latency fraction of DiTFastAttn in Diffusion Transformers compared with original attention mechanisms. The ASC technique reduces attention computation by 50%, with latency reduction slightly increasing as resolution increases. As resolution increases, WA-RS can reduce attention computation from 77% to 33%, and latency reduction ranges from 85% to 35%. The WA-RS and ASC techniques are orthogonal; they can be used simultaneously without additional overhead.

**Overall Latency of DiTFastAttn.** Fig. 8 shows the latency for image generation and attention as computation decreases when DiTFastAttn is applied. DiTFastAttn achieves end-to-end latency reduction for all three model at all compression ratio settings. PixArt-Sigma-2K shows the best performance, with overall generation latency at D6 being 56% of raw and overall attention latency at 37%. The result indicates that as resolution increases, DiTFastAttn achieves better performance in reducing latency for both overall attention and image generation.

## 4.5 Ablation Study

**DiTFastAttn Outperforms Single Methods**. As shown on the left of Fig. 9, DiTFastAttn maintains higher quality metrics compared to individual techniques with the same computation budget. Among single techniques, AST shows the best generation quality. However, beyond 2.2 FLOPs, further compression using AST significantly degrades the outputs, causing the search algorithm to terminate. DiTFastAttn supports further compression while maintaining better quality.

**Higher Steps Improve DiTFastAttn's Performance**. As shown on the middle of Fig. 9, we compared the performance of DiTFastAttn at different steps. It is evident that as the step increases, DiTFastAttn can compress more computation while maintaining quality.

**The Residual Caching Technique is Essential in Maintaining the Performance**. As shown on the right of Fig. 9, Window Attention with Residual Sharing maintains better generative performance

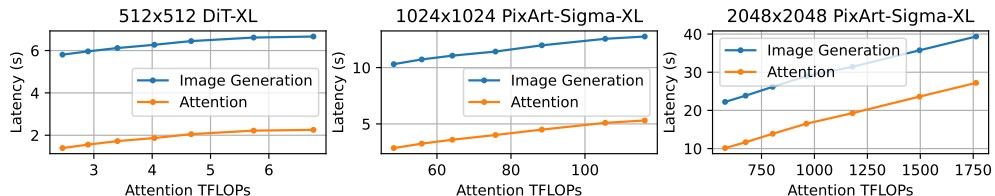

Figure 8: Latency of different resolutions of image generation under different compression ratios. DiT runs with a batch size of 8, while PixArt-Sigma models with a batch size of 1. The blue line delineates the latency for end-to-end image generation, whereas the orange line represents the latency of multi-head attention module.

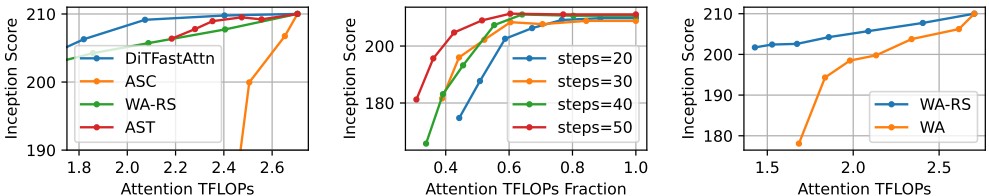

Figure 9: Ablation study on DiT-XL-2-512. Examination of methodological impact (Left), timesteps variability (Middle), and residual sharing (Right). 'WA' denotes Window Attention without the Residual Share (RS).

than Window Attention at the same compression ratio. Without residuals, window attention results in a significant performance drop.

## 5    Conclusion

In this paper, we introduce a novel post-training compression method, DiTFastAttention, to accelerate diffusion models. We identify three types of redundancy : (1) Redundancy in the spatial dimension. (2) Similarity between the neighboring steps in attention outputs. (3) Similarity between the conditional and unconditional inference in attention outputs. And we propose corresponding compression techniques: (1) Window Attention with Residual Sharing, (2) Attention Sharing across Timesteps, (3) Attention Sharing across CFG. The experiments show that DiTFastAttention significantly reduces the cost of attention and accelerates computation speeds.

**Limitations.** First, our method is a post-training compression technique and therefore cannot take advantage of training to avoid the performance drop. Second, our method mainly focuses on inference acceleration instead of VRAM reduction. When AST is applied, the attention hidden states from previous timestep will be stored and will bring extra VRAM usage. Third, our simple compression method may not find the optimal compression plan. Fourth, our method only reduces the cost of attention module.

## Acknowledgements

This work was supported by National Natural Science Foundation of China (No. 62325405, 62104128, U19B2019, U21B2031, 61832007, 62204164), Tsinghua EE Xilinx AI Research Fund, and Beijing National Research Center for Information Science and Technology (BNRist).

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

# A Appendix

## A.1 Societal Impacts

DiTFastAttention can enable more efficient deployment of diffusion transformer models, which have shown remarkable image and video generation capabilities. On the positive side, accelerating these models could democratize access to powerful generative AI by reducing computational requirements, and allowing broader adoption for creative and educational applications. However, there are also potential negative societal impacts that must be carefully considered. Highly realistic synthetic media could be exploited to create deepfakes for misinformation, fraud, or non-consensual editing. There are also potential privacy risks if generative models can reconstruct personal information from data. While DiTFastAttention does not inherently increase or reduce these risks compared to the original models, widening access makes misuse by malicious actors more likely. Therefore, safeguards against misuse and ethical guidelines for the responsible release of compressed models may be needed as this technology develops.

## A.2 Details of the Method for Deciding the Compression Plan

This algorithm has a computation complexity of $\mathcal{O}(|\mathcal{S}| \times |T| \times |M|^2 \times s^2)$, where $|\mathcal{S}|$ is the number of compression strategies (4 in our case), $|T|$ is the number of denoising steps, $|M|$ is the number of transformer layers, and $s$ is the sequence length. While the inference time for generating an image using the DiT has a computation complexity of $\mathcal{O}(|T| \times |M| \times s^2)$. Therefore, the greedy algorithm takes about $|\mathcal{S}| \times |M|$ of the image generation time. For example, the inference time for generating a 512×512 image using DiT-XL-2-512 is approximately 2 seconds, so the greedy algorithm takes around 224 seconds (2s × 28 transformer layers × 4 method candidates) to decide the compression plan, which is a reasonable overhead compared to the overall inference time.

The mean relative absolute error is a metric used to evaluate the performance of a model by measuring the average relative deviation between the outputs $O'$ and the raw outputs $O$. It is calculated as follows:

$$L(O, O') = \frac{1}{|O|_1} \sum_i \text{clip} \left( \frac{|O_i - O'_i|}{\max(|O_i|, |O'_i|) + \epsilon}, 0, 10 \right)$$

In this equation, $|O|_1$ represents the number of elements in the raw output vector $O$. The summation iterates over each element $i$ in the vectors $O$ and $O'$. For each element, the absolute difference between the raw output $O_i$ and the output $O'_i$ is calculated. This difference is then divided by the maximum value between $|O_i|$ and $|O'_i|$, which serves as a normalization factor to make the error relative to the magnitude of the output values. To avoid numerical instability in cases where both $O_i$ and $O'_i$ are very small or zero, a small positive constant $\epsilon$ (set to $10^{-6}$ in our experiments) is added to the denominator. The clip function ensures that the resulting ratio is clipped to the range [0, 10], preventing extreme values from dominating the overall error. The clipped ratios are summed and then divided by the total number of elements $|O|_1$ to obtain the mean relative absolute error. This metric provides a normalized measure of the average relative deviation between the predicted and raw outputs, with values ranging from 0 to 10 (maximum allowed error).

## A.3 Results for Video Generation

As shown in Fig. 10, the subjective evaluation of DiTFastAttn's application to video generation tasks revealed a significant performance distinction across configurations D1 through D6. Configurations D1 to D3 demonstrated effective performance, balancing computational efficiency with the retention of visual quality in generated videos. The subjective assessment indicated that videos generated under these configurations were smooth, with natural transitions between frames and preserved details that are critical for video quality. The maintenance of these qualities suggests that the model was able to effectively leverage the redundancies identified and apply the compression techniques without noticeable loss to the viewer.

In contrast, configurations D4 to D6, which applied more aggressive compression techniques, resulted in a noticeable deviation from the original video characteristics. The generated videos under D4 to D6 were still smooth and coherent, allowing them to represent the intended narrative or prompt with reasonable accuracy. This suggests that while aggressive compression can compromise certain

aspects of video quality, it can still be effective in scenarios where computational resources are limited and a high level of detail is not paramount.

The subjective results underscore the importance of finding an optimal balance between computational efficiency and generation quality when applying DiTFastAttn to video generation tasks. While configurations D1 to D3 offer a promising trade-off, the deviation observed in D4 to D6 highlights the need for careful consideration of the compression parameters. For practical deployment, it is crucial to select a DiTFastAttn configuration that aligns with the specific requirements of the application in terms of both performance and output quality.

## A.4 Latency Values in Different Settings

Table 3: Latency, FID and IS upon 50-step DPM-Solver. Attn Latency means the latency of self-attention. DiT-XL-2 runs with a batch size of 8. 50000 images used to generate FID and IS score

| Model | Resolution | Config | Latency (s) | Attn Latency (s) | FID | IS |
|-------|-----------|--------|-------------|------------------|-----|-----|
| DiT-XL-2 | 512×512 | Raw | 6.66 | 2.26 | 25.43 | 408.16 |
| | | D1 | 6.61 | 2.22 | 25.32 | 412.24 |
| | | D2 | 6.45 | 2.05 | 24.67 | 412.18 |
| | | D3 | 6.27 | 1.87 | 23.76 | 411.74 |
| | | D4 | 6.12 | 1.72 | 21.52 | 391.80 |
| | | D5 | 5.96 | 1.56 | 19.32 | 370.07 |
| | | D6 | 5.81 | 1.39 | 16.80 | 352.20 |

Table 4: Latency, FID and IS upon 50-step DPM-Solver. Attn Latency means the latency of self-attention. Models run with a batch size of 1. 30000 images used to generate FID, IS, and CLIP score

| Model | Config | Latency (s) | Attn Latency (s) | FID | IS | CLIP |
|-------|--------|-------------|------------------|-----|-----|------|
| PixArt-Sigma-XL 1024×1024 | Raw | 12.76 | 5.30 | 55.65 | 24.33 | 31.27 |
| | D1 | 12.55 | 5.10 | 55.73 | 24.27 | 31.27 |
| | D2 | 11.98 | 4.49 | 55.69 | 24.25 | 31.26 |
| | D3 | 11.42 | 4.01 | 55.61 | 24.16 | 31.25 |
| | D4 | 11.06 | 3.60 | 55.32 | 24.07 | 31.24 |
| | D5 | 10.73 | 3.25 | 54.54 | 24.17 | 31.22 |
| | D6 | 10.31 | 2.85 | 52.74 | 23.94 | 31.18 |
| PixArt-Sigma-XL 2048×2048 | Raw | 39.86 | 27.57 | 51.89 | 23.67 | 31.47 |
| | D1 | 35.75 | 23.62 | 52.34 | 23.28 | 31.46 |
| | D2 | 31.44 | 19.29 | 53.01 | 22.90 | 31.32 |
| | D3 | 28.99 | 16.51 | 52.54 | 22.96 | 31.36 |
| | D4 | 26.18 | 13.88 | 51.74 | 22.95 | 31.39 |
| | D5 | 23.86 | 11.66 | 51.22 | 22.82 | 31.34 |
| | D6 | 22.27 | 10.13 | 49.34 | 22.38 | 31.28 |

Table 5: Latency, FID and IS under DiT paper experiment setting (250-step IDDPM solver, cfg scale = 1.5). Attn Latency means the latency of self-attention. DiT-XL-2 runs with a batch size of 12

| Model | Resolution | Config | Latency (s) | Attn Latency (s) | FID | IS |
|-------|-----------|--------|-------------|------------------|-----|-----|
| DiT-XL-2 | 512×512 | Raw | 32.62 | 11.40 | 3.16 | 219.97 |
| | | D1 | 31.53 | 10.21 | 3.09 | 218.20 |
| | | D2 | 29.35 | 8.09 | 3.10 | 210.36 |
| | | D3 | 27.80 | 6.56 | 3.54 | 196.05 |
| | | D4 | 26.96 | 5.77 | 4.52 | 180.34 |

### A.5 Compression Plan

Fig. 11, 12, 13 display compression plan obtained after our greedy search method in different model settings as heatmaps. Each block stands for one layer at specific step. Both models exhibit three different kinds of redundancy, but the distribution of these redundancies across time steps and layers is quite different. The results indicate that there is no uniform compression plan for different DiT models and a search plan is essential in this case.

### A.6 Search Time

Table 6 show the plan search time in different configuration. Our greedy search method will try method that can achieve high compression ratio so plan search time is inversely proportional to threshold.

Table 6: Compression plan search time for three models

| Model | Resolution | Config | Plan Search Time |
|---|---|---|---|
| DiT-XL-2 | 512×512 | Raw | 04m39s |
| | | D2 | 04m08s |
| | | D4 | 03m49s |
| | | D6 | 03m14s |
| PixArt-Sigma-XL | 1024×1024 | Raw | 22m02s |
| | | D2 | 20m12s |
| | | D4 | 17m50s |
| | | D6 | 15m49s |
| PixArt-Sigma-XL | 2048×2048 | Raw | 1h50m13s |
| | | D2 | 1h46m04s |
| | | D4 | 1h22m53s |
| | | D6 | 1h23m01s |

### A.7 Metrics for Compression Plan Search

When designing the compression plan, we have considered to use other metrics including LPIPS and SSIM, and finally chose the existing metric mainly because of the speed of computation. We tested different SSIM compression schemes and found that when SSIM is chosen as the metric, to ensure the quality of the images generated, the threshold should be set at a small value of about 1/10 of the existing metric. We checked LPIPS as an alternative metric by decoding the hidden states into RGB space then calculate LPIPS on RGB space. We found that LPIPS is very insensitive to value changes in our use cases. Only small value changes can be observed when switch diffrent methods and always suggest to use sharing across timestep when threshold is set to 0.005 or smaller. Additionally, it takes a long time to compute LPIPS. In this way, we believe LPIPS is not a suitable metric for compression plan search.

### A.8 Negative Conditioning

Negative conditioning is a technique that widely used to improve generation quality by specifying what to exclude from the generated images. We explored the the impact of negative conditioning on our method using general negative prompt like 'Low quality' on PixArt-Sigma-XL. In the case, we found our method can preserve the effect of negative prompt on the generated images as shown in Fig. 14.

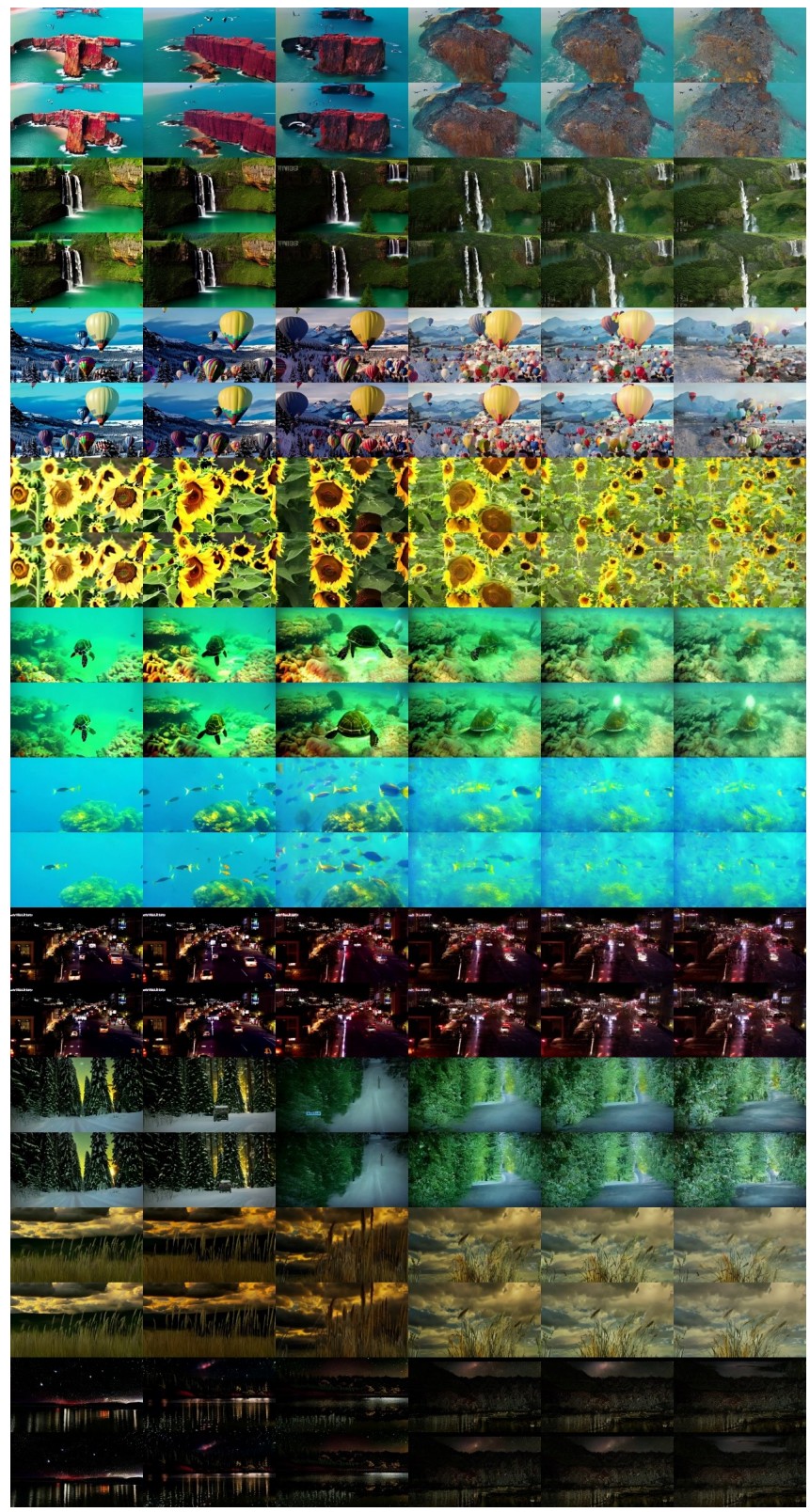

Figure 10: Comparison of video generation using OpenSora V1.1 at 240p resolution with 16 frames. The left column displays the original video, and the right columns illustrate the outputs from the D1 to D6 configuration.

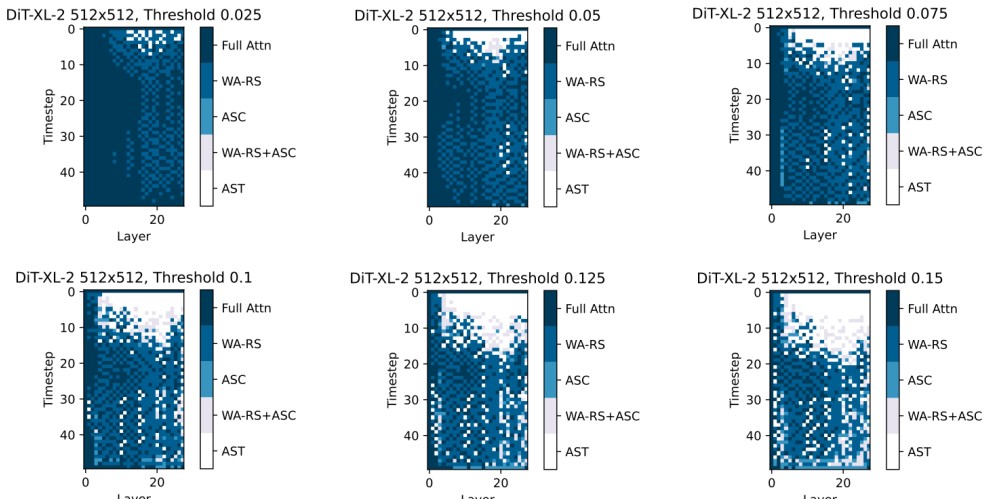

Figure 11: Compression plan for DiT-XL-2-512×512 at different thresholds with DPM solver step set to 50

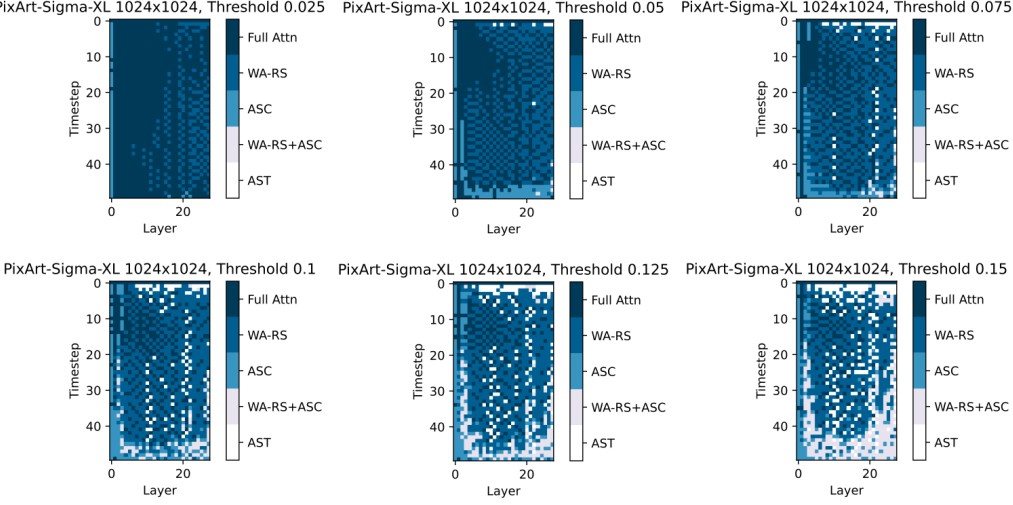

Figure 12: Compression plan for PixArt-Sigma-XL-1024×1024 at different thresholds with DPM solver step set to 50

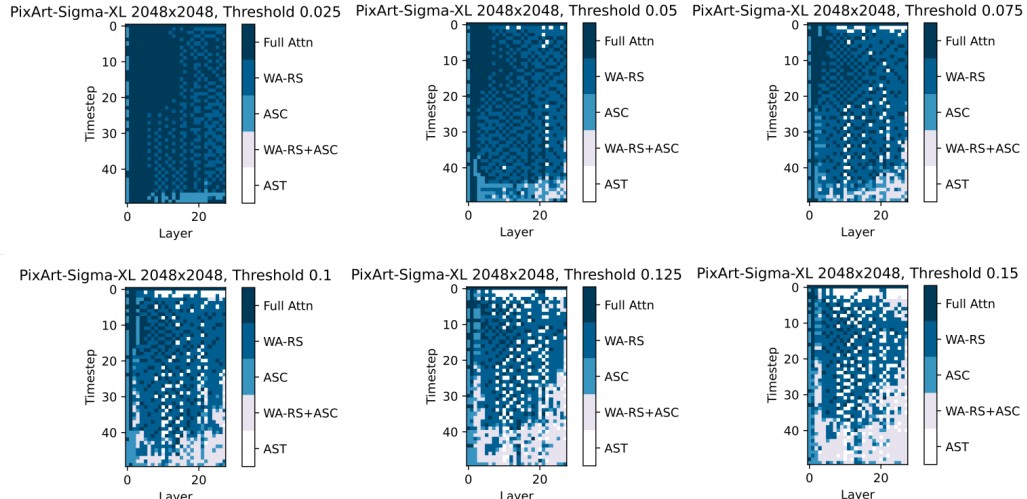

Figure 13: Compression plan for PixArt-Sigma-XL-2048×2048 at different thresholds with DPM solver step set to 50

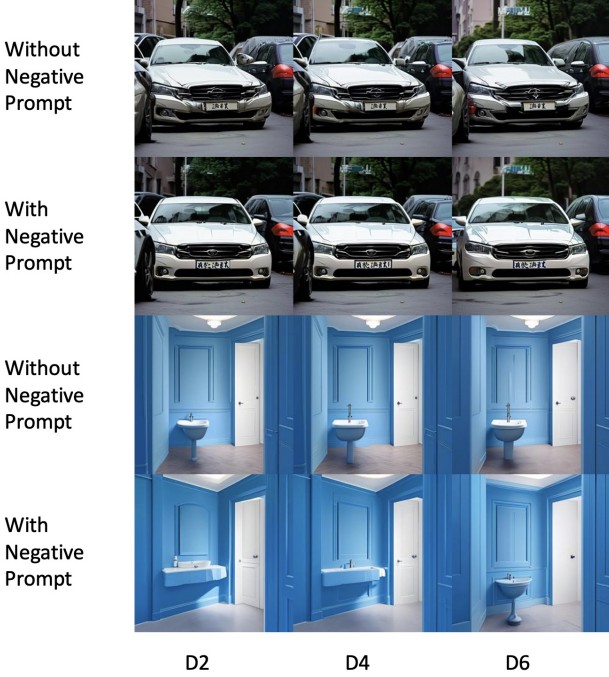

Figure 14: images generated by PixArt-Sigma-XL-1024 at different thresholds with/without negative prompt

