# OpenReview forum: "DiTFastAttn: Attention Compression for Diffusion Transformer Models"
_NeurIPS.cc/2024/Conference — NeurIPS 2024 poster_

### Official Review · Reviewer_8Fw6 · 2024-06-20

**Soundness:** 3
**Presentation:** 3
**Contribution:** 3
**Rating:** 5
**Confidence:** 4

**Summary:**

In this paper, the authors introduce a novel post-training model compression method aimed at accelerating Diffusion Transformers (DiT) used for image and video generation tasks. They identify the spatial, temporal, and conditional redundancy in attention blocks and propose the corresponding method to tackle them.

**Strengths:**

+ The proposed methods are interesting and effective.
+The writing is easy to understand.

**Weaknesses:**

- The authors sample 5K images to evaluate the generation quality while the standard setting is sampling 50K images (following original DiT).
- What is the adopted compression strategy in each layer and each timestep? It would be better to illustrate the searched compression plan in a figure.
- Can the proposed method be combined with other acceleration methods \eg Deepcache?
- Why do the authors only demonstrate the Inception Score in Figure 9? FID is the most common metric in image generation.
- What are the FLOPs and Latency in Table 1 exactly? The authors should demonstrate the value rather than only the fraction since there is still enough space.

**Questions:**

See weakness

**Limitations:**

See weakness

---

> ### Author Rebuttal · Authors · 2024-08-07
>
> We truly appreciate the reviewer for the insightful and constructive comments.
>
> > W1. The authors sample 5K images to evaluate the generation quality while the standard setting is sampling 50K images (following original DiT).
>
> Thanks. Following your suggestion, we have increased the evaluation size of ImageNet to 50K and COCO to 30K. Tables can be found in the global rebuttal (Table 1 & 2). More discussions can be found in the global rebuttal Q1.
>
> > W2. What is the adopted compression strategy in each layer and each timestep? It would be better to illustrate the searched compression plan in a figure.
>
> Thanks for your suggestion. We generated figures for adopted compression strategy in different threshold settings, check Q2 in global rebuttal and Figure 1 in the rebuttal PDF.
>
> > W3. Can the proposed method be combined with other acceleration methods \eg Deepcache?
>
> We believed that Deepcache benefits from the hierarchical nature of U-Nets so it can not be directly applied on DiT model. However, we do think our method can be combined with other acceleration methods. For example, our method is orthogonal with quantization method, so we can apply both quantization and our method to the model to achieve further acceleration.
>
> > W4. Why do the authors only demonstrate the Inception Score in Figure 9? FID is the most common metric in image generation.
>
> Showing only one metric in a figure tends to be more clear. In a future version of our paper, we will include an ablation study plot to further demonstrate the FID results.
>
> > W5. What are the FLOPs and Latency in Table 1 exactly? The authors should demonstrate the value rather than only the fraction since there is still enough space.
>
> Thanks. We make a summary table in the global rebuttal Q1 (Table 1 & 2) to contain exactly FLOPS and Latency. Summary tables like that will be included in the future version of our paper.

---

> > ### Comment · Reviewer_8Fw6 · 2024-08-08
> >
> > How much time do we need to search the compression strategy?

---

> ### Author Response · Authors · 2024-08-08
> **Compression strategy search time**
>
> Thanks for your comment.
>
> Search time varies for different models and parameter settings. We use a greedy method in compression plan search so increase threshold (more compression) will result in a shorter search time. The maximum search time for a model can be determined by setting the threshold to 0.  One thing to note is that once the search is completed, the compression strategy will be locally cached, eliminating the need for a repeated search. The cached strategy will be utilized the next time our method is applied for image or video generation. For reference, here we post the search times for DiT and PixArt-Sigma 1K using the default settings.
>
> #### Table1. Search time for DiT-XL-2-512
>
> | Threshold | Search Time |
> |-----------|-------------|
> | 0         | 04m39s      |
> | 0.05      | 04m08s      |
> | 0.1       | 03m49s      |
> | 0.15      | 03m14s      |
>
> #### Table2. Search time for PixArt-Sigma-XL-2-1024-MS
>
> | Threshold | Search Time |
> |-----------|-------------|
> | 0         | 22m02s      |
> | 0.05      | 20m12s      |
> | 0.1       | 17m50s      |
> | 0.15      | 15m49s      |

---

> ### Author Response · Authors · 2024-08-13
>
> Thanks for your patience. Here we post the results for DiT after changing our experiment setting to align with DiT paper. From this table, we observe that when the threshold is increased from 0 to 0.10, the IS drops from 210 to 180, and the FID decreases from 3.16 to 3.09, then increases to 4.52. The behavior of the FID in different settings follows the pattern reported in the previous discussion[1]. We observe a higher reduction in FLOPs and Attention FLOPs in this setting. We believe the current results demonstrate the effectiveness of our method across different settings.
>
> | Threshold | IS@50K | FID@50K | macs | attn_mac |
> |-----------|--------|---------|------|----------|
> | Raw setting | 219.9721 | 3.16   | 262359 | 33823   |
> | 0.025     | 218.1955 | 3.09   | 236265 | 18041   |
> | 0.050     | 210.3559 | 3.10   | 218865 | 12339   |
> | 0.075     | 196.05  | 3.54   | 203420 | 8777    |
> | 0.100     | 180.34  | 4.52   | 195682 | 7137    |
>
> So far, we increased evaluation sample size, changed experiments settings and provided results that aligned with the original paper according to W1; provided compression plan and search time that asked in W2 and FLOPs number that asked in W4; explained the combination with other acceleration model for W3; revised our paper according to W5. We are happy to answer additional questions and provide more clarification if needed. We kindly ask you to reconsider the score since we nearly reached the end of the discussion period.
>
> [1] Jayasumana S, Ramalingam S, Veit A, et al. Rethinking fid: Towards a better evaluation metric for image generation[C]// CVPR. 2024: 9307-9315.

---

### Official Review · Reviewer_5A6V · 2024-07-09

**Soundness:** 3
**Presentation:** 3
**Contribution:** 3
**Rating:** 7
**Confidence:** 4

**Summary:**

The authors observe computational redundancies across the three main dimensions of the generation process in DiTs -- space, sampling time, and conditional vs. unconditional forward passes due to CFG. Based on these observations, the authors introduce a set of approaches leveraging them to enable more efficient inference with negligible quality loss for pre-trained DiTs.

**Strengths:**

- This paper addresses an important avenue of research, as diffusion models are inherently very inefficient (as compared to, e.g., GANs) for inference, due to having to evaluate large models for a multitude of steps. While previous works such as DeepCache (Ma et al., 2023) or Cache Me If You Can (Wimbauer et al., 2023) have already investigated this problem in general, they do so specifically for diffusion U-Nets and many of these methods are not directly applicable to DiTs that do not share the hierarchical nature of U-Nets. Given the recent rise in popularity of DiT-based diffusion backbones for successful large-scale diffusion models (PixArt-alpha, Sora, Stable Diffusion 3, ...), providing methods that can successfully speed up the inference of these models is important. The authors present a method that is able to substantially reduce the cost incurred from evaluating attention, which makes up a large fraction of the FLOPS in high-resolution settings, in these diffusion transformers during inference while retaining most of the quality of the generated images, addressing this problem well.
The authors investigate and identify a set of redundancies during the generation process in diffusion transformers that relate to evaluating the attention mechanism. These findings are then used to motivate the presented methods.
- Going beyond naive approaches to exploit the presented redundancies, the authors introduce further, non-obvious tricks such as residual sharing to make them work well.
- The main method is thoroughly evaluated on two image diffusion models where it shows promising results.

**Weaknesses:**

- While the method as a whole is evaluated reasonably well, the evaluation of the effect of parts of the method & design choices seems lacking: there are missing ablations regarding different variants (e.g., naive approaches) of leveraging specific redundancies, and especially the evaluation of the compression plan algorithm is lacking and its hyperparameters (threshold $\delta$, number of samples).
- Open questions remain, especially regarding unorthodox design choices and whether the CFG redundancy is actually an attention redundancy as described or a general redundancy already thoroughly described in previous works. See the questions below.

**Questions:**

Questions regarding the main contributions of the paper
- While their approach to only computing attention locally does seem to be effective, I am surprised by the fact that the authors chose to apply a 1D window attention on the flattened image token sequence instead of one of the many 2D-aware local attention mechanisms such as (Shifting) Window Attention. These should have been the obvious choice in this situation, and the authors fail to motivate their unorthodox choice. While the attention map patterns presented in Fig. 3a do look somewhat like the effective attention pattern given by 1D window attention, I would presume that the additional diagonal lines next to the main diagonal correspond to the same location, but offset by one row. Besides generally being a more intuitive choice, the aforementioned approaches should also correspond to this observed pattern significantly better.
- Is it possible that the observed CFG redundancy is not an attention-specific aspect but a general one that is already well-described in previous literature? I would be very interested in a comparison with CFG not being applied at all for low noise timesteps, given the large range of prior works that found it beneficial to, substantially anneal the CFG scale for low noise timesteps (MDT, Gao et al., 2023; Analysis of Classifier-Free Guidance Weight Schedulers, Wang et al., 2024; and a wide range of non-published sources such as https://enzokro.dev/blog/posts/2022-11-28-sd-v2-schedules-1/, https://x.com/jeremyphoward/status/1584771100378288129, https://github.com/mcmonkeyprojects/sd-dynamic-thresholding), or even completely deactivate it (Applying Guidance in a Limited Interval Improves Sample and Distribution Quality in Diffusion Models, Karras et al., 2024).
- What is the distribution of which acceleration method is used how often, and where in the generation process?

Additional small questions
- Can the authors provide an explanation regarding the intuition of FID improving when reducing the attention FLOPS in Fig. 5?
- Subjectively, more compression results in a loss of contrast, both for generated images (albeit slightly), as well as videos. Do the authors have an explanation as to what might be causing this?
- For their similarity analyses in Fig. 4, the authors only analyze the cosine similarity of attention outputs. Are the magnitudes similar as well? Wouldn't this be a requirement to enable successful caching?

Presentation suggestions
- Fig. 5 is hard to parse due to the multitude and different scales of the presented metrics. I'd suggest that the authors separate the metrics into separate stacked graphs, which should also render these graphs more accessible for people with color perception impairments. I would also suggest adding arrows to indicate whether lower or higher is better for each individual metric. Similar improvements to other figures could also help make the paper more accessible.
- For a camera-ready version, I think the paper would benefit from additional uncurated qualitative examples

**Limitations:**

The authors have adequately addressed the limitations and societal impact of their work.

---

> ### Author Rebuttal · Authors · 2024-08-07
>
> We appreciate the reviewer for the insightful and constructive comments.
>
> > W1.  the evaluation of the effect of parts of the method & design choices seems lacking: there are missing ablations regarding different variants (e.g., naive approaches) of leveraging specific redundancies, and especially the evaluation of the compression plan algorithm is lacking and its hyperparameters (threshold $\delta$, number of samples).
>
> We show the effect of some naive approaches in Figure 9 in our paper, and we have added more images generated  (like Figure 5a in the attached pdf) and compression plan (like Figure 5c in the attached pdf) by naive approaches. We conducted an ablation study on number of samples, and we found that adding number of samples had little effect on the compression plan.
>
> DiT n_samples ablation
>
> | num of samples | Threshold | IS          | FID          |
> |---------|-----------|-------------|----------|
> | 8       | 0.05      | 206.43    | 29.97        |
> | 8       | 0.1       | 200.30    | 26.91        |
> | 8       | 0.15      | 179.13    | 23.11        |
> | 16      | 0.05      | 205.76    | 30.33        |
> | 16      | 0.1       | 197.16    | 27.62        |
> | 16      | 0.15      | 180.68    | 23.24         |
> | 32      | 0.05      | 205.63     | 30.35        |
> | 32      | 0.1       | 202.73    | 26.97         |
> | 32      | 0.15      | 180.22    | 24.55        |
>
> In global rebuttal Q5, we added more ablation studies.
>
> > W2. & Q2. Open questions remain, especially regarding unorthodox design choices and whether the CFG redundancy is actually an attention redundancy as described or a general redundancy already thoroughly described in previous works.
>
> Thanks for your comments. We believe that the issue of conditional guidance (CFG) redundancy is not limited to the attention mechanism alone. There has been some prior work exploring methods to reduce CFG redundancy. However, our approach differs from previous efforts, as we have examined the use of CFG sharing at a finer granularity. Specifically, we have assessed the influence of conditional and unconditional computations across different layers and different timesteps.
>
> Our work applies CFG sharing selectively, only on certain layers and time steps, in order to avoid significantly degrading the model's performance. In this way, our work acknowledges and builds upon existing research, while offering a novel perspective on addressing redundancy within the unique framework of diffusion models.
> Going forward, we plan to extend our method to other model components in the future.
>
> > Q1. While their approach to only computing attention locally does seem to be effective, I am surprised by the fact that the authors chose to apply a 1D window attention on the flattened image token sequence instead of one of the many 2D-aware local attention mechanisms such as (Shifting) Window Attention. These should have been the obvious choice in this situation, and the authors fail to motivate their unorthodox choice.
>
> Please see global rebuttal Q4.
>
> > Q3. What is the distribution of which acceleration method is used how often, and where in the generation process?
>
> Please check Q2 in global rebuttal and Figure 1 in the rebuttal pdf attached.
>
> > Q4. Can the authors provide an explanation regarding the intuition of FID improving when reducing the attention FLOPS in Fig. 5?
>
> We have thoroughly reviewed the evaluation code and confirmed that there are no issues on our end. We believe the discrepancy may be due to the inaccuracy of the FID metric, as indicated in previous research [1].
> The FID does not fully capture the visual quality of the generated images. However, changes in the FID score can still be meaningful, as they reflect shifts in the distribution of the generated images.
>
> [1] Jayasumana S, Ramalingam S, Veit A, et al. Rethinking fid: Towards a better evaluation metric for image generation[C]// CVPR. 2024: 9307-9315.
>
> > Q5. Subjectively, more compression results in a loss of contrast, both for generated images (albeit slightly), as well as videos. Do the authors have an explanation as to what might be causing this?
>
> We evaluated the images generated using different methods and found that the high threshold of ASC was likely the cause. We included an example generated by different methods and at varying ASC thresholds in the rebuttal PDF (Figure 5). We suspect that excessive CFG sharing can lead to a loss of detail and less pronounced edges in the generated images. If the trade-off between contract quality and ASC is a significant concern, the user should consider disabling the ASC method.
>
> > Q6. For their similarity analyses in Fig. 4, the authors only analyze the cosine similarity of attention outputs. Are the magnitudes similar as well? Wouldn't this be a requirement to enable successful caching?
>
> Thanks. We plot the magnitude of attention output, the plot can be found in the rebuttal pdf (Figure 6). It is similar to cosine similarity.
>
> > Q7. Fig. 5 is hard to parse due to the multitude and different scales of the presented metrics. I'd suggest that the authors separate the metrics into separate stacked graphs, which should also render these graphs more accessible for people with color perception impairments. I would also suggest adding arrows to indicate whether lower or higher is better for each individual metric. Similar improvements to other figures could also help make the paper more accessible.
>
> Thanks for your advice. Please see Q3 in global rebuttal.
>
> > Q8. For a camera-ready version, I think the paper would benefit from additional uncurated qualitative examples
>
> Thanks. We plan to include more uncurated qualitative examples, ablation studies, plots, and tables in the future version of our paper, including the results presented in the rebuttal PDF.

---

> > ### Comment · Reviewer_5A6V · 2024-08-12
> >
> > Thank you for the extensive response and for running the additional ablations in the short rebuttal timespan!
> >
> > Regarding your response to my CFG sharing comment, I'd like to clarify that I was primarily referring to the practice of omitting CFG for the final few steps altogether, which also seems to have a negligible effect in practice, and is substantially simpler.
> >
> > Regarding the Natten comparison, I'm really surprised that the Natten kernel is seemingly so inefficient despite the substantially smaller number of elements, and I thank the authors for providing that valuable context. I'd also suggest including that in a future revised version of the paper and potentially still running practical quality evaluations with Natten, as it should be substantially more FLOP-efficient and might catch up w.r.t. practical speed with innovations like flex attention. For now, the provided context is sufficient for me.

---

> > > ### Author Response · Authors · 2024-08-13
> > >
> > > Thanks for your comments and valuable suggestions.
> > >
> > > > Regarding your response to my CFG sharing comment, I'd like to clarify that I was primarily referring to the practice of omitting CFG for the final few steps altogether, which also seems to have a negligible effect in practice, and is substantially simpler.
> > >
> > > Thanks for your clarification. We conducted an ablation study on CFG dropping in the final steps. The results are shown in the following table. We observed that the method of CFG dropping in the final steps is effective in DiT. We also observed that our DiTFastAttn can work well with this method. We will add this to the appendix in our revised version of the paper, along with more evaluation results and generated images.
> > >
> > > | Setting             | IS@5K | FID@5K  |
> > > |---------------------|----------|------|
> > > | Raw                 | 208.3134 | 31.65|
> > > | CFG Dropping final 10% steps       | 278.4405 | 30.5 |
> > > | Threshold = 0.1     | 200.2979 | 26.91|
> > > | Threshold = 0.1 + CFG Dropping final 10% steps    | 262.5407 | 26.77|
> > >
> > > Note that the method only focuses on CFG dropping in the final steps, while our CFG sharing in our method reduces redundancy in other steps. Since the CFG dropping is manually set now, it is an interesting idea to automatically search for where to drop the CFG. We are considering researching this point in the future.
> > >
> > > > I'd also suggest including that in a future revised version of the paper and potentially still running practical quality evaluations with Natten, as it should be substantially more FLOP-efficient and might catch up w.r.t. practical speed with innovations like flex attention. For now, the provided context is sufficient for me.
> > >
> > > Thanks for your suggestion. We will include the Natten results in the revised version of our paper.
> > >
> > > So far, we have provided the results of 1D vs 2D local attention, n_samples, loss of contrast, magnitude, and the compression plan for Q1, 3, 5, 6, and 7. We have also provided a reasonable explanation for Q2, and an analysis of CFG sharing to address W2 and Q2. We are open to further discussion and happy to answer any additional questions. As we are nearing the end of the discussion period, we kindly ask you to reconsider the score.

---

> > > > ### Comment · Reviewer_5A6V · 2024-08-13
> > > >
> > > > Thank you for running the additional evaluation with CFG dropping. I'm surprised all the efficiency improvements actually improve the FID, which is counterintuitive. I'd have appreciated additional FLOP/MAC numbers to put these values into perspective, as my comment was primarily concerned with the question of whether just dropping CFG for the final few steps is a sufficient alternative for the proposed method, but looking at the MAC counts in the comment you posted in the main thread, I'd estimate the computational savings for your method at threshold 0.1 to be substantially greater than just CFG dropping. It's also interesting that CFG dropping apparently greatly boosts the IS. For me, this creates many interesting questions about the effect of CFG dropping that I wasn't thinking of before, but it has definitely addressed my concern about this work.
> > > >
> > > > Given that the authors have adequately addressed all my major concerns and questions and have addressed the question about the unusually high FID scores raised by the other reviewers, I will raise my score from 6 to 7 and hope for acceptance.

---

> > > > > ### Author Response · Authors · 2024-08-14
> > > > >
> > > > > Thank you kindly for this acknowledgement and recognition of our work.

---

### Official Review · Reviewer_e43K · 2024-07-12

**Soundness:** 2
**Presentation:** 3
**Contribution:** 3
**Rating:** 5
**Confidence:** 3

**Summary:**

The paper  presents a post-training model compression method aimed at reducing the computational complexity of Diffusion Transformers. The authors identify three key redundancies in the attention computation. To address these, they propose three techniques. The proposed methods compress the model FLOPs and enable more efficient deployment in resource-constrained environments. The paper includes extensive experimental results demonstrating significant reductions in computational cost and latency while maintaining generation quality.

**Strengths:**

- The identification and analysis of spatial, temporal, and conditional redundancies in attention mechanisms are thorough and well-founded.
- The authors conduct extensive experiments on multiple DiT models and various tasks, providing strong evidence of the effectiveness and generalizability of their methods.

**Weaknesses:**

- Limited Theoretical Analysis: I am in general agreement with the overall concept presented in the paper. However, the paper primarily focuses on empirical results, with limited theoretical analysis to support the proposed methods. A deeper theoretical foundation could help in understanding the generalizability and limitations of the techniques.
- Complexity  and Scalability: The paper does not extensively discuss the complexity of the compressed models. The figure 1 presents the efficacy under different resolutions. But it does not discuss the model complexity, running speed, and model size in these settings.
- Insufficient Comparisons: The paper does not compare the related methods, such as flashattention, and KV cache. This mainly weakens the contribution. A more detailed comparison with existing attention acceleration and model compression techniques would be beneficial.
- Vague Illustration：The authors present three key redundancy strategy. However, the paper lacks sufficient annotations and explanations and does not provide clear definitions for the three scenarios.
- Application: The paper mainly focuses on attention reduction, but It is not very effective at low resolutions because the attention mechanism accounts for a small proportion of the computational cost.

**Questions:**

- How do the authors ensure the generalizability of their method across different datasets and tasks? A more rigorous theoretical analysis could help understand the limitations and potential extensions of the proposed techniques.
- What is the model complexity, running speed, and model size under different compression settings?
- Can the authors offer clearer definitions and detailed illustrations for each redundancy type (spatial, temporal, and conditional redundancy)?

**Limitations:**

The authors have acknowledged several limitations of their work.

---

> ### Author Rebuttal · Authors · 2024-08-07
>
> We appreciate the reviewer for the insightful and constructive comments.
>
> > W1 & Q1. How do the authors ensure the generalizability of their method across different datasets and tasks? A more rigorous theoretical analysis could help understand the limitations and potential extensions of the proposed techniques.
>
> About generalizability, our experiments demonstrate the effectiveness of DiTFastAttn across multiple datasets and tasks:
> - Image generation: We evaluated on class conditional generation (for DiT) and text conditional generation (for PixArt-Sigma).
> - Video generation: We successfully applied DiTFastAttn to OpenSora for video generation tasks.
>
> Our method shows improved compression ratios and quality preservation as resolution increases from 512x512 to 2048x2048, indicating good scalability.
> This diversity in datasets, tasks, and model architectures provides strong evidence for the generalizability of our approach. However, we acknowledge that further evaluation on additional domains could further strengthen this claim.
>
> For theoretical analysis, we thank the reviewer for this valuable suggestion. While our current work focuses on empirical results, we agree that a more rigorous theoretical analysis would be beneficial.
> - We have provided rigorous computational complexity analysis for each proposed technique (WA-RS, AST, ASC) and their combinations.
> Some directions we could explore in future work include:
> - Investigate how the error propagates through the denoising steps when using approximated attention outputs.
> - Quantify the information loss when using window attention compared to full attention.
>
> We will consider adding a discussion of these theoretical aspects in an extended version of the paper.
>
>
> > W2 & Q2. What is the model complexity, running speed, and model size under different compression settings?
>
> In Appendix A.4 of our paper, we discussed the complexity of our compression method. Additionally, in the global rebuttal, we have added summary tables (Table 1 and 2) to present the exact FLOPs and latency (running speed). The result shows our method is effective to reduce both the complexity and improve the running speed. The model size will not change.
>
> For model size and VRAM usage, when AST or WA-RS is applied, the hidden states from the previous timestep are stored. The shape of stored hidden states of a layer is [batchsize*2, num_of_head, seqlength, hidden_size]. For example, with the DiT XL/2 512 model, at a batchsize of 1, the VRAM usage of the original model is 11652MB. To store the FP16 hidden states, an additional 2016MB of VRAM is needed, which is 17.3% of the original model's VRAM usage. We will include a VRAM analysis in the limitations section.
>
> > W3. Insufficient Comparisons: The paper does not compare the related methods, such as flashattention, and KV cache. This mainly weakens the contribution. A more detailed comparison with existing attention acceleration and model compression techniques would be beneficial.
>
> Thanks. Our baseline (we denoted as raw setting in our method) has used flashattention in their attention computation, and our method can achieve up to 1.6x speedup compared to it. We'll add a note to the paper to avoid confusion.
>
> KV cache is a method used in large language models for auto-regressive generation. In diffusion model, DiT don't need to attend the tokens from previous timesteps. KV cache isn't applicable to diffusion models like DiT. So we cannot compare our method with KV cache. Note that our method is inspired by caching concepts but applied differently:
> - Caching activation outputs across denoising timesteps and conditional/unconditional branches.
> - WA-RS technique caches residuals to maintain long-range dependencies with efficient local attention.
>
>
> > W4 & Q3. Can the authors offer clearer definitions and detailed illustrations for each redundancy type (spatial, temporal, and conditional redundancy)?
>
> Although we have provided some definitions in our paper, we agree that providing more clarity on these concepts will strengthen our paper. Here we denote $X$ as input, $Y$ as output
>
> Window Attention with Residual Sharing (WA-RS):
> - Definition: The original paper Eq.1 and Eq.2.
> - Illustration: The original paper Figure.3. We will add a heat map visualization of the attention matrix, clearly showing the concentration of attention values along the diagonal.
>
> Attention Sharing across Timesteps (AST):
> - Definition: For step k, $Y_k=W_{o}O_k$ and $Y_{k+1} = \\{Y_k$  if  $AST_k = 1$; $W_{o}O_{k+1}$ if $AST_k = 0\\}$
>
> - Illustration: The original paper Figure.2 is clear
> ﻿
> Attention Sharing across CFG (ASC):
> - Definition: For step k, $Q_{k} = \\{W_{Q}X_{k,:c/2}$ if $ASC_k = 1$, $W_{Q}X_{k}$ if $ASC_k = 0\\}$ and so as $K_{k}$ and $V_{k}$. The output is $Y_{k} = \\{[W_{o}O_k, W_{o}O_k]$ if $ASC_k = 1$; $W_{o}O_{k}$ if $ASC_k = 0\\}$
> - Illustration: The original paper Figure.2 is clear
>
> > W5. Application: The paper mainly focuses on attention reduction, but It is not very effective at low resolutions because the attention mechanism accounts for a small proportion of the computational cost.
>
> While it's true that our method provides greater benefits at higher resolutions, we believe this is actually a strength rather than a limitation:
> - Even at lower resolutions, our method still provides measurable improvements. For example, on the 512x512 DiT model, we achieve up to 31% FLOP reduction and 10% latency reduction for attention computation.
> - As transformer-based diffusion models continue to scale up in size and target higher resolutions, the relative importance of attention computation increases. Our method is thus well-positioned to provide even greater benefits for future large-scale models.

---

> > ### Author Response · Authors · 2024-08-13
> >
> > Dear Reviewer e43K, Thanks so much again for the time and effort in our work. According to the comments and concerns, we provided more analysis and further discuss the related points. We also conduct a variety of ablation studies and experiments that we showed in the Global rebuttal.
> >
> > As the rebuttal period is about to close, may I know if our rebuttal addresses the concerns? If there are further concerns or questions, we are welcome to address them. Thanks again for taking the time to review our work and provide insightful comments.

---

> > ### Comment · Reviewer_e43K · 2024-08-14
> >
> > After reviewing the authors’ rebuttal and all subsequent responses, I find that most of my concerns have been addressed. Consequently, I have decided to revise my rating to a positive one. I recommend that the additional experiments and analyses discussed be incorporated into the revised version of the manuscript.

---

### Official Review · Reviewer_CWF1 · 2024-07-12

**Soundness:** 2
**Presentation:** 3
**Contribution:** 3
**Rating:** 5
**Confidence:** 4

**Summary:**

This paper proposes a combination of novel techniques to reduce the self-attention computation in diffusion transformers (DiTs), without requiring finetuning. This methods leverage locality in attention scores, similarity in timesteps, and similarity with classifier free guidance. Overall, the proposed method is capable of maintaining image quality while significantly reducing self-attention FLOPS.

**Strengths:**

The paper proposes three novel approaches with compelling empirical justification for each. The methods were then ablated and demonstrate good visual performance while reducing computational overhead during inference. While the authors acknowledge limitations of their kernel implementation, their results may have a strong impact in both downstream deployment and in explainable AI for generative image models.

**Weaknesses:**

While the visual results are compelling, the paper has limited quantitative evaluations, and may be in a regime with low statistical SNR requiring additional samples. Furthermore, the authors focus on improving FLOPS, however, they do not discuss VRAM usage which would be significant for consumer grade devices and generating longer videos. Finally, there are issues with figure 5 which significantly hinders readability.

**Questions:**

1) You should cite Sora in the introduction and not OpenSora.
2) While the inclusion of residual caching is novel, how is your windowed attention method different from neighborhood attention (Natten)? This should be cited with local attention.
3) Including the 1-sigma boundaries (e.g. aggregated over 1k samples) in the MSE plots on Fig 3a would help understand how much variation there is for the attention layers. Additionally, is that DiT-XL or Pixart-Sigma? Identifying the model in the caption would be a good idea.
4) You explored the impact with CFG but have you looked at the impact with negative conditioning? Does sharing the conditional attention perform similarly or degrade?
5) There appears to be some interesting behavior in Figure 4, which may be interesting to explore from an explainability perspective.
6) For your compression plan search, did you consider using other metrics such as LPIPS or SSIM? Would they impact the chosen plan?
7) The use of 5k image samples for evaluation is unusual. Typically the evaluation is 50k for ImageNet and 30k for COCO, as a larger sample size will reduce statistical noise (FID can be very noisy below 15k samples).
8) When you evaluated FID, IS, CLIP, did you do so on the same 5k samples used for determining the plan? I would have used the ImageNet/COCO validation set for the plan search and then evaluated on the test set to ensure the two distributions were non-overlapping.
9) What dataset did you use for your OpenSora evaluation?
10) Did you consider looking at the impact on VRAM? If there is no impact, stating so in the limitations would be appropriate.
11) If increasing the step count improves the result in Fig 9, then this suggests that there is an optimal tradeoff between attention FLOPs and step count (iso-sampling FLOPS). Performing that analysis may strengthen adoption of your technique.
12) Reporting relative performance as percentages is difficult to read. It would be clearer to report them as ratios (e.g. 0.93 vs 93%)

Finally, Figure 5 is unreadable and requires revision.
- FID and IS should not be plotted on the same axis as they are very different in scale. Use a dual-axis plot if you want to combine them.
- The points should be connected with a line like in Figure 8 and 9.
- I would only plot CLIP and FID (not IS) for Pixart, which will make the plot easier to read.
- Your CLIP scores appear too high, where they should be between 0.2 and 0.35.

Please provide a revision to this figure or a table of the data plotted therein.

**Limitations:**

The authors adequately discuss limitations of their approach, although should include a note about VRAM.

---

> ### Author Rebuttal · Authors · 2024-08-07
>
> We truly appreciate the reviewer for the insightful and constructive comments.
>
> > Q1. You should cite Sora in the introduction and not OpenSora.
>
> Thanks. We have added the citation of Sora in the introduction.
>
> > Q2. How is your windowed attention method different from neighborhood attention (Natten)? This should be cited with local attention.
>
> We adopted 1D window attention, which can be recognized as 1D neighborhood attention, and we have added references to Natten in the article.  The evaluation shows our 1D window attention is more efficient (See global rebuttal Q4).
>
> > Q3. Including the 1-sigma boundaries (e.g. aggregated over 1k samples) in the MSE plots on Fig 3a would help understand how much variation there is for the attention layers.
>
> We have added the MSE change plot on 1k samples and included the 1-sigma boundaries. We have also added the model type in the figure title and caption. The results indicate that the variation is not significant. Please refer to Figure 4.b in the attached rebuttal PDF.
>
> > Q4. You explored the impact with CFG but have you looked at the impact with negative conditioning? Does sharing the conditional attention perform similarly or degrade?
>
> We investigated the impact of negative conditioning. The attentions with positive conditioning and negative conditioning have a high degree of similarity, and our method is also effective in reducing computation with negative conditioning. We provided an example in the attached PDF (Figure 4c, "low quality" as the negative prompt). We also added a plot to show that similarity across CFG with negative prompt behave similar as the one without negative prompt (Figure 4d). We will add more examples and analysis of negative conditioning in our paper.
>
> > Q5. There appears to be some interesting behavior in Figure 4, which may be interesting to explore from an explainability perspective.
>
> In Figure 4a, we observe that the middle timesteps exhibit less similarity, which may suggest that the structural transformation occurs primarily in the middle timesteps, while the other timesteps focus on noise removal and refinement. In Figure 4b, we note that the middle layers show more differences between the conditional and unconditional cases, indicating that the middle layers are responsible for the condition-specific processing.
>
> > Q6. For your compression plan search, did you consider using other metrics such as LPIPS or SSIM? Would they impact the chosen plan?
>
> Yes, we've considered LPIPS and SSIM for our comparison. LPIPS only works for image comparisons and requires the channel number to be set to 3, whereas we are comparing latent space (with 4 channels). LPIPS also needs to use a network as a feature extractor, which results in a significant inference overhead.
>
> We also tried the SSIM metric, and the results are included in Table 4 of the global rebuttal. The SSIM metric appears to achieve comparable results to our own metric. However, SSIM is more computationally complex to calculate. Therefore, we have decided to use our own metric for this task. We have included the compression plan in Figure 2 of the rebuttal PDF. Interestingly, the SSIM metric selects more lower-level layers for compression, while the existing metric chooses to compress the higher layers. We will research this in the future.
>
> > Q7. Evaluation sample number
>
> Please see global rebuttal Q1. Our conclusions remain unchanged at the larger evaluation size.
>
> > Q8. When you evaluated FID, IS, CLIP, did you do so on the same 5k samples used for determining the plan? I would have used the ImageNet/COCO validation set for the plan search and then evaluated on the test set to ensure the two distributions were non-overlapping.
>
> For the calibration process, we only use class labels (for DiT) and text prompts (for PixArt-Sigma) to generate a small set of images (8 images for DiT, 6 images for PixArt 1K, and 4 for PixArt 2K). No real images are used in the calibration process, so there is no risk of overlapping.
>
> > Q9. What dataset did you use for your OpenSora evaluation?
>
> Since the video generation process is too resource-intensive, and we found the evaluation metrics to be unstable when the sample size is small, we did not apply these metrics to the generated videos. Instead, we included some sample generated examples. These example videos were generated using the default prompts provided by OpenSora.
>
> > Q10. Did you consider looking at the impact on VRAM? If there is no impact, stating so in the limitations would be appropriate.
>
> When AST or WA-RS is applied, the hidden states from the previous timestep are stored. The shape of stored hidden states of a layer is [batchsize*2, num_of_head, seqlength, hidden_size]. For example, with the DiT XL/2 512 model, at a batchsize of 1, the VRAM usage of the original model is 11652MB. To store the FP16 hidden states, an additional 2016MB of VRAM is needed, which is 17.3% of the original model's VRAM usage. We will include a VRAM analysis in the limitations section.
>
> > Q11. If increasing the step count improves the result in Fig 9, then this suggests that there is an optimal tradeoff between attention FLOPs and step count (iso-sampling FLOPS).
>
> In the rebuttal PDF, we have added a FLOPs-IS Pareto front plot (Figure 4.a). Interestingly, we observed that different compression ratios require different optimal settings (steps and DiTFastAttn threshold). We will include this plot and the related discussion in the appendix of the future version of the paper.
>
> > Q12. Reporting relative performance as percentages is difficult to read. It would be clearer to report them as ratios
>
> Thanks. We've changed the format of relative performance from percentage to ratio.
>
> > Fig 5 & CLIP issues
>
> Figure 5 is now changed accordingly (check global rebuttal and Figure 3 in the pdf). For CLIP, we use the package torchmetric to calculate it so the scale is different (0-100 vs 0-1). It is now changed to 0-1.

---

> > ### Comment · Reviewer_CWF1 · 2024-08-09
> >
> > I thank the authors for their effort with additional evaluations, the SSIM difference and compression plans are especially interesting.
> >
> > There are a few points that I would like clarification on.
> >
> > **Q1.**
> >
> > There appears to be an issue with your evaluation metrics. The FID scores are too high (DiT-XL/2 should be around 2.2 with CFG, and Pixart-sigma around 9). Similarly, your IS scores for DiT-XL are too high and too low for Pixart. Were you able to reproduce the metrics from the respective papers without your compression method?
> >
> > **Q2.**
> >
> > > LPIPS only works for image comparisons and requires the channel number to be set to 3...
> >
> > This is partially true, where it would be possible to retrain a vgg model as per [1] in the latent space. However, it would also be feasible to simply decode the latent images and apply LPIPS in RGB space (this is also a differentiable operation).
> >
> > **Q3.**
> >
> > > For the calibration process, we only use class labels (for DiT) and text prompts (for PixArt-Sigma) to generate a small set of images (8 images for DiT, 6 images for PixArt 1K, and 4 for PixArt 2K). No real images are used in the calibration process, so there is no risk of overlapping.
> >
> > Is this a sufficient number of images to establish a robust compression plan? If the goal is to find a method for downstream inference performance, I would want to have at least 1k samples (maybe more).
> >
> > **Q4.**
> >
> > Given the early focus on full self-attention in the compression plans, would some downscaled method like HiDiffusion [2] be applicable?
> >
> > **Q5.**
> >
> > > ...we have added a FLOPs-IS Pareto front plot...
> >
> > Interpreting this plot is difficult, perhaps adding marker shapes for each compression ratio would make it clearer. However, if I understand correctly, it suggests that using no compression at 20 steps performs similarly to using moderate compression at 30 steps? If so, can you give an example for where your approach would be preferable to simply reducing the sampling steps?
> >
> > [1] Zhang, R., et.al. (2018), "The Unreasonable Effectiveness of Deep Features as a Perceptual Metric"
> >
> > [2] Zhang, S., et.al. (2023), "HiDiffusion: Unlocking Higher-Resolution Creativity and Efficiency in Pretrained Diffusion Models"

---

> > > ### Comment · Reviewer_8Fw6 · 2024-08-13
> > >
> > > I also have the same question: 'The FID scores are too high (DiT-XL/2 should be around 2.2 with CFG, and Pixart-sigma around 9). Similarly, your IS scores for DiT-XL are too high and too low for Pixart.'

---

> ### Author Response · Authors · 2024-08-13
>
> Thanks for your additional questions. Here is our reply:
>
> > Q1. Evaluation metrics
>
> The experiment settings are different in the following ways:
> - 1. The number of timesteps. To compare DiT with ADM and LDM, DiT uses a relatively high number of diffusion timesteps (250). For most image generation applications, 20-50 steps would be reasonable. Other methods, such as SD-XL and PixArt series, also evaluate on 20-50 steps. That is why we use 20-50 steps in our paper.
> - 2. Evaluation software. We use pytorch-fid & torchmetric to calculate FID and IS scores, while DiT uses the TensorFlow evaluation suite from the ADM paper. This resulted in some differences in the FID and IS scores.
> - 3. We use a cfg_scale of 4, which is the default setting for both DiT official code and diffusers DiT pipeline. In the DiT paper, they use a cfg_scale of 1.5 to match the settings used for ADM and LDM.
>
> We have modified the code to reproduce the results. However, it is not feasible to resample the 50K images under the timestep=250 setting during the rebuttal period (this would take approximately 55 GPU hours). We will provide a partial set of the results soon.
>
> For PixArt-Sigma, they used a curated set of 30,000 images instead of COCO to calculate FID. Since the dataset is not publicly released, we cannot reproduce those results. We will clearly explain the experiment settings in the appendix.
>
> > Q2. LPIPS implementation
>
> Thank you for the suggestion. We have implemented LPIPS as an additional evaluation metric (by decoding the hidden states into RGB space). However, we found that LPIPS has problems for our use case:
> - 1. Insensitivity to Value Changes: We observed that LPIPS is quite insensitive to value changes in our experiments. Only small value changes were observed when switching between different methods, and LPIPS always suggested using sharing across timesteps when the threshold was set to 0.005 or smaller.
> - 2. Computational Overhead: Using LPIPS as a metric takes significantly more time. Therefore, we were unable to provide the LPIPS results within the rebuttal period due to the computational requirements.
>
> > Q3. Number of calibration images
>
> We conducted an ablation study on the number of samples generated during the plan search, and found that increasing the number of samples had little effect on the resulting compression plan.
>
> | n samples | threshold | IS    | FID           |
> |---------|-----------|-------|---------------|
> | 8       | 0.05      | 206.4319 | 29.97 |
> | 8       | 0.1       | 200.2979 | 26.91  |
> | 8       | 0.15      | 179.1331 | 23.114 |
> | 16      | 0.05      | 205.7642 | 30.33 |
> | 16      | 0.1       | 197.1631 | 27.62 |
> | 16      | 0.15      | 180.6786 | 23.24  |
> | 32      | 0.05      | 205.626  | 30.35 |
> | 32      | 0.1       | 202.7274 | 26.97 |
> | 32      | 0.15      | 180.2231 | 24.55 |
>
> From the results, the IS and FID did not change significantly as we varied the number of samples. Nevertheless, users can easily set 'n' to a larger number if desired, as it is a configurable parameter in our code.
>
> >Q4. downscaled method like HiDiffusion be applicable?
>
> We have carefully reviewed the HiDiffusion approach, which is designed for UNet-based diffusion models. We have considered how this method may be applicable to our framework:
> - 1. HiDiffusion proposed using local attention to replace global attention in the top layers of the UNet, as global attention in the upper blocks is computationally dominant. However, in DiT, the amount of self-attention computation is equal across each layer. Therefore, we do not need to focus solely on specific layers, and instead use a search-based method to automatically identify which layers can be replaced.
> 2. HiDiffusion use the Modified Shifted Window Attention, where the window area is set differently across diffusion timesteps. This approach may be applicable to our method as well. However, as mentioned in the Global rebuttal (Q4), the inference speed of these attention mechanisms could be slower. It is also an interesting topic to research.
>
> >Q5. Pareto front question
>
> Thanks for your suggestion. We have followed your advice and added marker shapes. Here is our observation from the plot:
> - The point representing compression at 30 steps (yellow marker at (2.7 TFLOPs, 208 IS)) is superior to using no compression at 20 steps (blue marker at (2.9 TFLOPs, 207 IS)). This is because the compressed 30-step model has lower computational cost (TFLOPs) while achieving a higher Inception Score (IS).
> - For a high FLOPs budget (> 3.5T), it is better to use moderate compression with 50 steps.
> - For a medium FLOPs budget (2-3.5T), it is preferable to use slight compression with 20 steps or moderate compression with 30 steps.
> - For a low FLOPs budget (< 2T), the optimal choice would be to use high compression with either 20 or 30 steps.

---

> ### Author Response · Authors · 2024-08-13
>
> Thank you for your patience. Here we present the results for DiT after changing our experimental settings to align with the DiT paper.
> | Threshold | IS@50K | FID@50K | macs | attn_mac |
> |-----------|--------|---------|------|----------|
> | Raw | 219.9721 | 3.16   | 262359 | 33823   |
> | 0.025     | 218.1955 | 3.09   | 236265 | 18041   |
> | 0.050     | 210.3559 | 3.10   | 218865 | 12339   |
> | 0.075     | 196.05  | 3.54   | 203420 | 8777    |
> | 0.100     | 180.34  | 4.52   | 195682 | 7137    |
>
> From this table, we observe that when the threshold is increased from 0 to 0.10, the IS drops from 210 to 180, and the FID decreases from 3.16 to 3.09, then increases to 4.52. The behavior of the FID in different settings follows the pattern reported in the previous discussion[1]. We observe a higher reduction in FLOPs and Attention FLOPs in this setting. We believe the current results demonstrate the effectiveness of our method across different settings.
>
> So far, we provide explanations, revision and examples according to each concerns. We are open to more discussion and will to provide more clarification if needed. As we nearly reached the end of the discussion period, we kindly ask you to reconsider the score.
>
> [1] Jayasumana S, Ramalingam S, Veit A, et al. Rethinking fid: Towards a better evaluation metric for image generation[C]// CVPR. 2024: 9307-9315.

---

> > ### Comment · Reviewer_CWF1 · 2024-08-14
> >
> > Thank you for the follow-up clarification.
> >
> > Q1.
> >
> > The updated FID and IS scores are closer to what is expected. While I understand your desire to evaluate under more practical settings, maintaining a consistent evaluation protocol is necessary to compare with other works. This requirement for comparison is much more sensitive to image count and cfg scale rather than sampler, which is why many recent works deviate in sampler algorithm and step count (computational practicality).
> >
> > From your updated evaluations, I interpret the results as significant degradation beyond a threshold of 0.05, below which the speedup is not significant.
> >
> > Q2.
> >
> > The results you described with LPIPS are interesting, and should be included in the revision for completeness. If anything, they will at the very least indicate that a less computational method achieves similar or better results for less overhead.
> >
> > Q3.
> >
> > Your results exhibit a significant improvement using more samples for a threshold above 0.1. When interpreting FID, a shift of 1-2 points can be significant, although this may be reduced with the updated conditions from Q1.
> >
> > I remain concerned with the low number of images used in calibration, which I believe may serve to highlight issues in the generality of FID and IS as standard evaluation metrics. While I understand that more samples are computationally expensive, the cost largely becomes irrelevant if it need only be performed once per model.
> >
> >
> > ---
> >
> > I thank the authors for their effort in addressing my questions and those of the other reviewers. Additionally, I commend the authors on their detailed investigations, which make this an interesting technical work. As such, I am inclined to raise my score. However, I believe the variable quality as a function of calibration sample count brings into question the efficacy of the results. Furthermore, the results presented bring into question the appropriateness of using FID and IS as primary evaluation metrics rather than as a sanity check against significant degradation. Given the aforementioned pros and cons, I raise my score from 4 to 5.

---

> ### Author Response · Authors · 2024-08-14
> **Clarification on updated results**
>
> We appreciate your valuable feedback. In the revised version, we will include the LPIPS results and additional evaluation metrics, as well as more uncurated qualitative examples, as you suggested.
>
> Regarding your interpretation of the updated evaluations (Q1), there seems to be some misunderstanding. Our findings indicate that even at a threshold of 0.025, the method was able to significantly reduce the attention FLOPs by 46%, while maintaining a satisfactory generation effect. **At a threshold of 0.05, the attention FLOPs were reduced by 63% without significant degradation**. The results show that as the number of steps increases, the threshold required to achieve the same compression ratio decreases. For example, **with 250 steps, a threshold of 0.025 can achieve the same compression ratio as a threshold of 0.125 with 20 steps**. This means that for larger timesteps, a smaller threshold is needed to reach the same compression ratio.
>
> Therefore, we believe our results demonstrate that in the experiment setting of 250 steps, a small threshold can indeed achieve good compression performance. We will adjust the threshold values and include more results in the range of 0 to 0.05 to better illustrate this point.
>
> We kindly ask you to reconsider the score if we have addressed all of your concern.

---

> > ### Comment · Reviewer_CWF1 · 2024-08-14
> >
> > Thank you for the additional context.
> >
> > I remind the authors that FLOPs is not a good metric for inference speedup, especially considering that many inference workloads are memory bound. While FLOPs can serve as a heuristic, the bigger picture is more complicated. Instead, overall inference latency is a more appropriate metric of which a significant deviation is not observed for thresholds below 0.1. From the other results, it is my belief that your upper performance bound is limited by the number of calibration samples, which results in plans that are unable to effectively generalize across the standard evaluation metrics. The authors should focus on that aspect: reducing degradation for higher thresholds to achieve significant latency reductions, rather than more results with thresholds below 0.05.

---

> > > ### Author Response · Authors · 2024-08-14
> > >
> > > Thank you for your reminder. We have tested the latency to the table for your reference. We observed a 30% reduction in DiT-XL/2 attention latency at a threshold of 0.05 (timesteps=250, 512x512 generation), showing that our method not only reduces FLOPs but also results in a significant speedup in the attention calculation part. Regarding the overall inference latency, your observation is correct that the latency reduction fraction is lower than the attention latency reduction fraction. However, the proportion of attention computation in the overall computation would increase along with resolution (as shown in Figure 1 of our paper). Our method can significantly reduce the overall latency on the generation of high-resolution images. For 2K image generation, attention computation contributes more than 70% of the total computation time, so a 30% reduction in attention latency will result in more than a 21% reduction in overall latency.
> > >
> > > Regarding the number of calibration samples, we agree with your idea and will increase the number to further reduce degradation. However, it will take more computation cost. This is a trade-off between calibration time and performance. We will carefully evaluate the number of samples in the future. Thanks again for your remind and advice. If we have addressed all of your concern, we kindly ask you to reconsider the score.

---

### Author Rebuttal · Authors · 2024-08-06

Thanks all the reviewers for the time and effort taken to provide valuable insights and comments on our work. Here we provide experiment results and answers for some common questions and comments:

> 1. The use of 5k image samples for evaluation is not enough.

We have increased the evaluation size of ImageNet to 50K samples and COCO to 30K samples. The table below showing the results of DiT (Table 1) and PixArt-Sigma 1K (Table 2). Plots of the ImageNet@50K and COCO@30K results can be found in the rebuttal PDF (Figure 3). The results are more stable and demonstrate the same trend of change as the previous 5K results. These updated results will be included in the future version of the paper.

#### Table1. DiT ImageNet 50K Results

| Threshold | MACs (GFLOPs) | Attn_MAC (GFLOPs) | Latency | Attn_Latency | IS | FID |
|-----------|------:|----------:|---------:|--------:|---:|----:|
| 0         | 20989 | 2706      | 2.841s    | 0.890s  | 400.64 | 23.99 |
| 0.025     | 20623 | 2383      | 2.863s    | 0.918s  | 402.24 | 23.86 |
| 0.05      | 20105 | 2088      | 2.854s    | 0.914s   | 400.28 | 22.95 |
| 0.075     | 19560 | 1832      | 2.892s    | 0.909s   | 401.37 | 21.75 |
| 0.1       | 19032 | 1598      | 2.782s    | 0.828s   | 385.48 | 19.74 |
| 0.125     | 18432 | 1361      | 2.769s    | 0.799s   | 330.35 | 18.09 |
| 0.15      | 17796 | 1209      | 2.658s    | 0.710s   | 328.21 | 15.10 |

#### Table2. PixArt 1K COCO 30K Results

| Threshold | MACs(GFLOPs)  | Attn_MAC (GFLOPs)| Latency| Attn_Latency | IS | FID | CLIP    |
|----|----:|----:|-----:|----:|---:|---:|-----:|
| 0         | 132693    | 46464    | 2.437s    | 0.937s  | 24.25| 55.70| 0.31377 |
| 0.025     | 129478    | 43662    | 2.459s    | 0.958s  | 24.28| 55.67| 0.31378 |
| 0.05      | 123701    | 38385    | 2.508s    | 0.993s  | 24.23| 55.58| 0.31371 |
| 0.075     | 117661    | 33041    | 2.472s    | 0.962s  | 24.18| 55.28| 0.31365 |
| 0.1       | 113351    | 29427    | 2.417s    | 0.922s  | 23.98| 55.11| 0.31342 |
| 0.125     | 108101    | 25330    | 2.372s    | 0.868s  | 23.74| 53.90| 0.31342 |
| 0.15      | 103403    | 22131    | 2.300s    | 0.798s  | 23.40| 51.68| 0.31314 |

> 2. What is the adopted compression strategy?

We generated plots for the adopted compression strategies under different threshold settings. Please refer to Figure 1 in the rebuttal PDF. The results show that all the strategies (WA-RS, ASC, WA-RS+ASC, and AST) are well-distributed, demonstrating that our strategy search algorithm is effective. An interesting observation is that the different strategies tend to be concentrated in specific layers and timesteps.

> 3. Figure 5 is unreadable.

  We have optimized the display of Figure 5. We show the result of each metrics seperately. Arrows are added to indicate whether lower or higher is better for each individual metric. We show 4 plots in the rebuttal pdf (Figure 3) as an example due to limited spaces.

> 4. What kind of local attention is used? Why use 1d local attention instead of 2d local attention?

We use sliding window attention for WA-RS, which is the same as the sliding window attention used in Longformer and the 1D neighborhood attention. Our method can also be generalized to 2D local attention. However, we observe that 2D local attention is not efficient enough. This is because the GPU memory access of the K and V tensors is not sequential, resulting in extra data gathering overhead.
We evaluate the inference latency of adopting Natten 2D attention and our sliding window attention. We use a kernel size of 127 for the sliding window, and a kernel size of 5 for the Natten 2D attention. The latency results are shown in Table 3 below. Based on these results, we chose to use 1D local attention.

#### Table3. Latency of different kernel

| Kernel          | Attention Latency |
|-----------------|--------------------|
| Window Attention   | 0.828s              |
| Natten 2D Attention       | 0.926s              |

> 5. More ablations and examples are needed.

In the rebuttal PDF,
   1. we conducted additional ablation studies on different metrics used in the search (Figure 2). We tested different SSIM compression schemes and found that when SSIM is chosen as the metric, to ensure the quality of the images generated, the threshold should be set at a small value of about 1/10 of the existing metric.
  2. We investigated different  the impact of the negative prompt (Figure 4c) and found that negative prompt remain effective with our method.
  3. We checked the effect of CFG sharing on contrast (Figure 5). Loss of contrast can happen when using a large threshold for ASC.
  4. We provided examples of the Pareto front of number of steps vs. TFLOPS (Figure 4a), which can served as guidance for user to choose appropriate compression setting.
  5. We provided the MSE plot of 1K samples with 1-sigma in DiT (Figure 4b) to show that window part have a higher MSE and larger variation across samples.
  6. We plotted the Magnitude of Attention Outputs Across Step and CFG Dimensions in DiT (Figure 6) to show magnitude of attention output show a similar result with cosine similarity.

#### Table4. SSIM vs Current Metric

| Threshold | Metric | FID@5K    | IS@5K   | MACs (GFLOPs)| Attn_MAC (GFLOPs)| Latency | Attn_Latency |
|---|----|------|------|----|------|------|-----|
| 0.005     | SSIM   | 28.77 | 201.92 | 19324 | 1770    | 2.851s   | 0.862s   |
| 0.01      | SSIM   | 26.71 | 200.92 | 18611 | 1492    | 2.794s   | 0.825s   |
| 0.015     | SSIM   | 25.44 | 192.42 | 18042 | 1286    | 2.764s   | 0.774s   |
| 0.05      | Current   | 30.33 | 205.76 | 20102 | 2064    | 2.957s   | 0.950s   |
| 0.1       | Current   | 27.62 | 197.16 | 18976 | 1579    | 2.834s   | 0.851s   |
| 0.15      | Current   | 23.24  | 180.68 | 18028 | 1229    | 2.713s   | 0.756s   |

---

> ### Comment · Reviewer_8Fw6 · 2024-08-10
>
> The results in Table 1 and Table 2 are strange.  Lower IS and lower FID? I think that when the quality of the image decreases, the IS will be lower but FID will be higher.

---

> > ### Author Response · Authors · 2024-08-10
> >
> > Thank you for pointing out this observation.
> > We have thoroughly reviewed the evaluation code and confirmed that there are no issues on our end. We believe the discrepancy may be due to the inaccuracy of the FID metric, as indicated in previous research [1]. The FID does not fully capture the visual quality of the generated images. However, changes in the FID score can still be meaningful, as they reflect shifts in the distribution of the generated images.
> >
> > We checked a large number of the generated images, and found that the quality does decrease with an increase in the threshold. Based on this finding, we will add more generated samples and human evaluation into the paper.
> >
> > [1] Jayasumana S, Ramalingam S, Veit A, et al. Rethinking fid: Towards a better evaluation metric for image generation[C]// CVPR. 2024: 9307-9315.

---

### Author Response · Authors · 2024-08-13
**Additional reply for the concerns about evaluation setting**

## Explanation
The experiment settings are different in the following ways:
1. The number of timesteps. To compare DiT with ADM and LDM, DiT uses a relatively high number of diffusion timesteps (250). For most image generation applications, 20-50 steps would be reasonable. Other methods, such as SD-XL and PixArt series, also evaluate on 20-50 steps. That is why we use 20-50 steps in our paper.
2. Evaluation software. We use pytorch-fid & torchmetric to calculate FID and IS scores, while DiT uses the TensorFlow evaluation suite from the ADM paper. This resulted in some differences in the FID and IS scores.
3. We use a cfg_scale of 4, which is the default setting for both DiT official code and diffusers DiT pipeline. In the DiT paper, they use a cfg_scale of 1.5 to match the settings used for ADM and LDM.

## Results after changing experiment setting to align with DiT paper

| Threshold | IS@50K | FID@50K | macs | attn_mac |
|-----------|--------|---------|------|----------|
| Raw setting | 219.9721 | 3.16   | 262359 | 33823   |
| 0.025     | 218.1955 | 3.09   | 236265 | 18041   |
| 0.050     | 210.3559 | 3.10   | 218865 | 12339   |
| 0.075     | 196.05  | 3.54   | 203420 | 8777    |
| 0.100     | 180.34  | 4.52   | 195682 | 7137    |

## Analysis of results

From this table, we observe that when the threshold is increased from 0 to 0.10, the IS drops from 210 to 180, and the FID decreases from 3.16 to 3.09, then increases to 4.52. The behavior of the FID in different settings follows the pattern reported in the previous discussion[1]. We observe a higher reduction in FLOPs and Attention FLOPs in this setting. We believe the current results demonstrate the effectiveness of our method across different settings.

[1] Jayasumana S, Ramalingam S, Veit A, et al. Rethinking fid: Towards a better evaluation metric for image generation[C]// CVPR. 2024: 9307-9315.

---

> ### Author Response · Authors · 2024-08-14
> **Clarification on updated results**
>
> Our findings indicate that even at a threshold of 0.025, the method was able to significantly reduce the attention FLOPs by 46%, while maintaining a satisfactory generation effect. **At a threshold of 0.05, the attention FLOPs were reduced by 63% without significant degradation**. The results show that as the number of steps increases, the threshold required to achieve the same compression ratio decreases. For example, **with 250 steps, a threshold of 0.025 can achieve the same compression ratio as a threshold of 0.125 with 20 steps**. This means that for larger timesteps, a smaller threshold is needed to reach the same compression ratio.
>
> Therefore, we believe our results demonstrate that in the experiment setting of 250 steps, a small threshold can indeed achieve good compression performance. We will adjust the threshold values and include more results in the range of 0 to 0.05 to better illustrate this point.

---

### Decision · Program_Chairs · 2024-09-25

**Decision:**

Accept (poster)

**Comment:**

The paper proposed a combination of techniques to reduce the self-attention computation in diffusion transformers (DiTs), without requiring finetuning. The proposed method leverages locality in attention scores, similarity in timesteps, and similarity with classifier free guidance. Overall, the proposed method is capable of maintaining image quality while significantly reducing self-attention FLOPS. After rebuttal and discussions, the recommendations from the reviewers were 3XBorderline Accept and 1XAccept. The reviewers were consistent in the paper's contributions while they also made lots of comments in improveing the paper.